# More Than One Teacher: Adaptive Multi-Guidance Policy Optimization for Diverse Exploration

## Abstract

Reinforcement Learning with Verifiable Rewards (RLVR) is a promising paradigm for enhancing the reasoning ability in Large Language Models (LLMs). However, prevailing methods primarily rely on self-exploration or a single off-policy teacher to elicit long chain-of-thought (LongCoT) reasoning, which may introduce intrinsic model biases and restrict exploration, ultimately limiting reasoning diversity and performance. Grounded in the principle of the "Wisdom of Crowds" and drawing inspiration from multi-teacher strategies in knowledge distillation, we introduce **A**daptive **M**ulti-Guidance **P**olicy **O**ptimization (**AMPO**), a novel framework that adaptively leverages guidance from multiple proficient teacher models, but only when the on-policy model fails to generate correct solutions. This "guidance-on-demand" approach expands exploration while preserving the value of self-discovery. Moreover, AMPO incorporates a comprehension-based selection mechanism, prompting the student to learn from the reasoning paths that it is most likely to comprehend, thus balancing broad exploration with effective exploitation. Extensive experiments show AMPO substantially outperforms a strong baseline (GRPO), with a **4.3**% improvement on mathematical reasoning tasks and **10.9%** on out-of-distribution tasks, while significantly boosting Pass@k performance and enabling more diverse exploration. Notably, using four peer-sized teachers, our method outperforms a single, more powerful teacher (*e.g.*, DeepSeek-R1) under the same data budget and achieves comparable results to approaches trained with significantly more data. These results demonstrate a more efficient and scalable path to superior reasoning and generalizability. *Our code is available at* `https://anonymous.4open.science/r/7fBQd46C`, *which will be made public after double-blind review.*

## 1 Introduction

Recent advances in long chain-of-thought (LongCoT) Jaech et al. (2024); Guo et al. (2025); Team et al. (2025) have endowed Large Language Models (LLMs) with remarkable complex reasoning capabilities. A key driver behind this progress is the Reinforcement Learning with Verifiable Rewards (RLVR) paradigm. By enabling models to explore a vast solution space and learn from feedback on self-generated outputs, RLVR facilitates the development of robust reasoning abilities through trial and error, offering a more effective learning pathway than Supervised Fine-Tuning (SFT) Wei et al. (2021); Shi et al. (2025); Chu et al. (2025), especially in complex domains like mathematical.

Despite its success, RLVR implemented with on-policy algorithms like Group Relative Policy Optimization (GRPO) Shao et al. (2024) faces a fundamental challenge: the model's exploration is confined within its own knowledge boundaries Yue et al. (2025); Gandhi et al. (2025). While it can refine existing skills, it struggles to acquire new knowledge or reasoning strategies that are far beyond its initial capabilities. This limitation is exacerbated by the "capacity-difficulty mismatch", where models consistently fail on complex problems, leading to sparse rewards and training instability Yu et al. (2025a); Liu et al. (2025e).

To overcome this limitation, recent work has explored mixed-policy reinforcement learning (Mixed-Policy RL). For instance, integrating off-policy reasoning traces from a single, capable teacher

model into on-policy RL Yan et al. (2025), or interleaving RL with SFT in high-quality demonstrations Ma et al. (2025). While effective, relying on a single source, no matter how powerful, imposes a critical bottleneck: it might constrain the student to a homogeneous reasoning style and introduce intrinsic teacher biases (Tian et al., 2025; Xu et al., 2025). This "single-source limitation" restricts the breadth of exploration, standing in stark contrast to the established success of multi-teacher strategies in knowledge distillation Beyer et al. (2022), which have been shown to enrich data diversity and improve model robustness.

Building on this insight, we introduce **A**daptive **M**ulti-Guidance **P**olicy **O**ptimization (**AMPO**), a novel Mixed-Policy RL framework grounded in the principle of the **"Wisdom of Crowds"**, the idea that the collective judgment of diverse peers often surpasses that of a single expert Leimeister (2010); Kameda et al. (2022). Instead of relying on a single proprietary giant (*e.g.*, GPT4o or DeepSeek-R1), AMPO leverages the collective intelligence of multiple **peer-sized** models. This approach not only democratizes access to high-quality guidance but also ensures a broader coverage of the reasoning space. AMPO operates on a "guidance-on-demand" principle: external guidance from diverse teachers replaces on-policy failures only when the student model is unable to solve a problem, thus maximizing the value of self-exploration. Furthermore, AMPO employs a comprehension-based guidance selection mechanism. Inspired by recent work Yu et al. (2025b) on probability-based rewards, this mechanism guides the student to learn from the reasoning trace it is most likely to comprehend, striking a deliberate balance between broad exploration and effective exploitation.

We validate AMPO across six in-distribution (mathematics) and four out-of-distribution benchmarks based on Qwen2.5-7B-Ins Team (2024). Our results demonstrate that AMPO substantially outperforms GRPO, with an average improvement of **4.3**% on math benchmarks and a striking **10.9%** on out-of-distribution tasks. Most notably, our experiments provide two-fold evidence for the superiority of diverse peer guidance over a single expert source. First, under the same data budget, using four peer-sized teachers outperforms a single stronger teacher (DeepSeek-R1). Second, regarding data efficiency, AMPO achieves performance comparable to LUFFY (Yan et al., 2025), a strong single-teacher baseline, despite using nearly $5\times$ less data (8.5k vs. 46k). This empirically validates that diverse peer guidance yields superior learning signals. Further analysis confirms that AMPO enhances Pass@k performance and maintains higher training entropy, fostering a superior balance between exploration and exploitation. Consistent superiority over GRPO in experiments with smaller model and other family model further demonstrates the broad effectiveness of our method. We further explored the impact of guidance replacement count and the composition of the Multi-Guidance Pool on our approach, thereby introducing new avenues for future research. Our main contributions can be summarized as follows:

- We propose **AMPO**, a novel Mixed-Policy RL framework that leverages multiple peer-sized teacher guidance to unlock the "Wisdom of Crowds", effectively mitigating the potential learning diversity issues inherent in single-teacher approaches.

- We design an **Adaptive Multi-Guidance Replacement** strategy that minimizes intervention by providing external guidance only upon complete on-policy failure, preserving self-discovery while enhancing reasoning efficiency.

- We introduce a **Comprehension-based Guidance Selection** mechanism that improves learning effectiveness by guiding the model to assimilate the most comprehensible external solutions, demonstrably boosting performance.

- Extensive experiments show that our method significantly outperforms strong baselines in both performance and generalization. Crucially, we demonstrate that a diverse team of peer models is more effective than a single strong expert, establishing a more efficient and scalable path for improving LLM reasoning.

## 2 RELATED WORKS

**Knowledge Distillation in LLM Reasoning.** Following the emergence of powerful models like OpenAI-o1 Jaech et al. (2024) and DeepSeek-R1 Guo et al. (2025), research on enhancing LLM reasoning has surged. One line of research uses knowledge distillation Beyer et al. (2022) to transfer reasoning capability from stronger large language models(LLMs) to smaller ones, thereby reducing deployment costs. Models like OpenR1 Hugging Face (2025), OpenThought Guha et al. (2025),

and AM Zhao et al. (2025) were created by distilling extensive demonstration data from a powerful teacher (*e.g.*, DeepSeek-R1). Critically, recent studies Tian et al. (2025); Xu et al. (2025) argue that relying on a single teacher will limit the model's learning perspectives, reduce its problem-solving diversity. Instead, they propose multi-teacher strategies to aggregate diverse reasoning paths, effectively leveraging the "Wisdom of Crowds" or collective intelligence Leimeister (2010); Kameda et al. (2022) to enhance robustness and generalization. Nevertheless, knowledge distillation via SFT is often criticized for promoting memorization over a genuine understanding of complex thought patterns Chu et al. (2025), leading to suboptimal performance beyond the training distribution.

**Reinforcement Learning with Verifiable Rewards.** Recent research has shifted focus to Reinforcement Learning with Verifiable Rewards (RLVR). This paradigm incentivizes the model to autonomously develop its reasoning processes by rewarding correct final answers. Studies show that RLVR enables models to cultivate advanced cognitive skills, substantially enhancing their reasoning abilities Shao et al. (2024); Hu et al. (2025). However, such a on-policy RL confines models to their inherent knowledge boundaries, primarily amplifying existing capabilities rather than acquiring new ones Yue et al. (2025); Gandhi et al. (2025). Concurrently, the "capacity-difficulty mismatch" remains a challenge, where a lack of rewards for difficult problems leads to unstable training, as highlighted by works Yu et al. (2025a); Liu et al. (2025e).

**Mixed-Policy Reinforcement Learning.** To help models transcend their capability limits and mitigate training instability from sparse rewards, integrating off-policy expert data into on-policy RL has emerged as a promising direction. To inject external knowledge, current methods often resort to heuristic mixing strategies. For instance, mixing expert data into on-policy rollout responses Yan et al. (2025), or interleaving RL with SFT in high-quality demonstrations (Ma et al., 2025). Alternative methods use external guidance as prompts Liu et al. (2025b); Wu et al. (2025) or incorporate SFT objective as auxiliary losses within the RL framework Fu et al. (2025); Zhang et al. (2025). However, these approaches have two key limitations: 1) they primarily rely on a single teacher, which could restrict the diversity of learning; 2) they often integrate data in a static way, without considering the needs or comprehension of the model. In contrast, our proposed AMPO framework directly addresses both limitations by replacing the single-teacher paradigm with a Multi-Guidance Pool to enhance exploratory diversity, and substituting static data integration with an adaptive, comprehension-based mechanism that provides guidance only when the model truly needs it.

# 3 ADAPTIVE MULTI-GUIDANCE POLICY OPTIMIZATION

## 3.1 GROUP RELATIVE POLICY OPTIMIZATION (GRPO)

GRPO Shao et al. (2024) has demonstrated exceptional performance across various tasks, particularly within the RLVR paradigm. Its key innovation lies in estimating advantages by normalizing reward scores within a group of solutions sampled for a given query. This approach eliminates the need for an additional value model, which is typically required by traditional RL algorithms like Proximal Policy Optimization (PPO) Schulman et al. (2017), thereby significantly reducing computational resources and enhancing training efficiency.

Specifically, let $\pi_{\theta_{\text{old}}}$ denote the policy model before an update and $\pi_\theta$ be the policy model being updated. Given a query $q$ and a reward function $R(\cdot)$, $\pi_{\theta_{\text{old}}}$ generates $G$ responses $\{o_1, \ldots, o_G\}$, the advantage $A_{i,t}$ in GRPO is computed as the normalized reward within the group:

$$A_{i,t} = \frac{R(o_i) - \text{mean}\left(\{R(o_i) \mid o_i \sim \pi_{\theta_{\text{old}}}(\cdot|q), i = 1, \ldots, G\}\right)}{\text{std}\left(\{R(o_i) \mid o_i \sim \pi_{\theta_{\text{old}}}(\cdot|q), i = 1, \ldots, G\}\right)}, \quad (1)$$

where $R(\cdot)$ is typically a rule-based verifier that returns a binary reward based on correctness in practical GRPO applications.

The optimization objective for GRPO, as implemented in our work, is then defined as:

$$\mathcal{J}_{\text{GRPO}}(\pi_\theta) = \frac{1}{G} \sum_{i=1}^{G} \frac{1}{|o_i|} \sum_{t=1}^{|o_i|} \left\{ \min\left[r_{i,t} A_{i,t}, \text{clip}(r_{i,t}, 1 - \epsilon, 1 + \epsilon) A_{i,t}\right] \right\}, \quad (2)$$

where the importance sampling term $r_{i,t} = \frac{\pi_\theta(o_{i,t}|q, o_{i,<t})}{\pi_{\theta_{\text{old}}}(o_{i,t}|q, o_{i,<t})}$ calibrates the gradient discrepancy arising from on-policy updates Sutton et al. (1999), as solutions are generated by $\pi_{\theta_{\text{old}}}$ rather than

$\pi_\theta$. The clip function serves to maintain the stability of policy updates. Following recent works Yu et al. (2025a); Yan et al. (2025), we omit the KL divergence term.

## 3.2 EXTENDING GRPO WITH ADAPTIVE MULTI-GUIDANCE REPLACEMENT

Based on the findings from multi-teacher knowledge distillation Tian et al. (2025); Bin et al. (2024); Xu et al. (2025), we hypothesized that introducing diverse off-policy guidance can foster more varied exploration and lead to superior training outcomes. Consequently, we construct a **Multi-Guidance Pool** ($\mathcal{P}_G$), which contains correct off-policy responses from multiple distinct teacher models. Drawing inspiration from recent research Liu et al. (2025e), we implement an **Adaptive Multi-Guidance Replacement** mechanism that provides external guidance only when sparse rewards occur. The overall learning pipeline is illustrated in Figure 1.

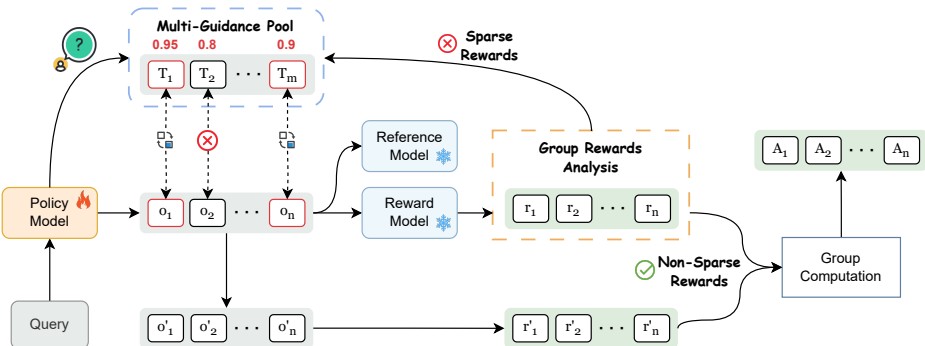

Figure 1: The AMPO training framework. It enhances exploration by adaptively replacing on-policy failures with external solutions from a Multi-Guidance Pool only when sparse rewards occur. The selection of external guidance is prioritized based on the Policy Model's comprehension score for each option, ensuring effective learning.

Specifically, given a query $q$, the policy model $\pi_{\theta_{old}}$ first generates G responses. These are then evaluated by the reward function $R(\cdot)$. If all responses receive a reward below a predefined threshold $\tau$, we deem it a sparse reward scenario and set the replacement flag $I$ to True:

$$I = \begin{cases} \text{True} & \text{if } R(o_i) < \tau, \ \forall i \in \{1, \dots, G\} \text{ where } o_i \sim \pi_{\theta_{old}}(\cdot|q) \\ \text{False} & \text{otherwise} \end{cases}. \quad (3)$$

This ensures that replacement is triggered only if none of the on-policy responses are fully correct. Subsequently, if $I = \text{True}$, AMPO will randomly select $k$ of the erroneous on-policy responses for replacement. These are substituted with the top-$k$ off-policy responses selected from $\mathcal{P}_G$ via our comprehension-based strategy, which is detailed in Section 3.3. Here, $k = \min(k_0, N_g)$, where $k_0$ is the target number of replacements, and $N_g$ is the number of available off-policy responses for query $q$ in the pool. This process forms an augmented batch $\mathcal{G}_{aug}$:

$$\mathcal{G}_{aug} = \begin{cases} \{o_i \sim \pi_{\theta_{old}}(\cdot|q), i = 1, \dots, N_{on}\} \cup \{o_j \in \mathcal{P}_G, j = 1, \dots, N_{off}\} & \text{if } I = \text{True} \\ \{o_i \sim \pi_{\theta_{old}}(\cdot|q), i = 1, \dots, G\} & \text{otherwise} \end{cases}, \quad (4)$$

where $N_{on} = G - k$, $N_{off} = k$ denote the counts of remaining on-policy and selected off-policy responses, respectively. Through this mechanism, the model is assured of learning from correct solutions in every update step, while still prioritizing its own self-discovered paths. External guidance is activated only when the model faces challenges beyond its current capabilities, enhancing its ability to acquire new knowledge for difficult problems.

## 3.3 COMPREHENSION-BASED GUIDANCE SELECTION

After an adaptive replacement is triggered, the challenge becomes selecting the most valuable guidance from the multiple off-policy responses in the Multi-Guidance Pool ($\mathcal{P}_G$). To address this, we

need a metric to quantify how "understandable" a teacher's reasoning is to the student model. Inspired by the Probability-based Reward mechanism Yu et al. (2025b), we adopt the student model's likelihood of generating the correct answer, given a teacher's reasoning path, as a proxy for comprehension. We term this metric the **Probability Reward ($r_p$)**.

To compute this, let an off-policy response from the pool be $o^{\text{off}} = (z^{\text{off}}, y)$, comprising the teacher's reasoning path $z^{\text{off}}$ and its final answer $y$. We formulate a corrected trajectory $o^* = (z^{\text{off}}, y^*)$ by replacing $y$ with the ground-truth answer $y^*$. The comprehension score $r_p$ for this trajectory is the geometric mean probability of on-policy model ($\pi_\theta$) generating the correct answer tokens in $y^*$, conditioned on the $z^{\text{off}}$. This is calculated using average log-probabilities and clipped to $[0, 1]$:

$$r_p(o^{\text{off}}) = \text{clip}(\exp(\frac{1}{|y^*|} \sum_{\tau_i \in y^*} \log \pi_\theta(\tau_i | z^{\text{off}}, y^*_{<i})), 0, 1). \tag{5}$$

A higher $r_p$ score indicates that the reasoning of the teacher is more aligned with the student's internal knowledge representation, making it a more suitable learning target. By ranking all available guidance in $\mathcal{P}_G$ for a given query and selecting the top-$k$ responses with the highest $r_p$ scores, our **Comprehension-based Guidance Selection** strategy ensures the model learns from the most effective and assimilable examples.

Notably, our reward function $R(\cdot)$ includes a format reward to incentivize placing answers within special tags as described in Appendix A.2.3, consistent with the extraction of $y$ needed for the $r_p$ calculation Yu et al. (2025b). In cases where multiple guidance options achieve the same top $r_p$ score, we use response length as a tie-breaker, prioritizing shorter, more concise reasoning paths.

## 3.4 POLICY OPTIMIZATION WITH MULTI-GUIDANCE

During the policy update, we employ a mixed-objective approach for the augmented batch $\mathcal{G}_{\text{aug}}$. First, a unified advantage $\hat{A}_{k,t}$ is computed for each response $o_k \in \mathcal{G}_{\text{aug}}$ by normalizing rewards across the entire batch:

$$\hat{A}_{k,t} = \frac{R(o_k) - \text{mean}(\{R(o_k) \mid o_k \in \mathcal{G}_{\text{aug}}, k = 1, \ldots, |\mathcal{G}_{\text{aug}}|\})}{\text{std}(\{R(o_k) \mid o_k \in \mathcal{G}_{\text{aug}}, k = 1, \ldots, |\mathcal{G}_{\text{aug}}|\})}. \tag{6}$$

To ensure clarity, we denote the advantages derived from on-policy and off-policy responses as $\hat{A}^{\text{on}}_{i,t}$ and $\hat{A}^{\text{off}}_{j,t}$, respectively. For each off-policy response $o^{\text{off}}_j$, we assume its originating teacher policy as $\pi_{\phi_j}(\cdot) \approx 1$ for computational efficiency to bypass teacher log-probabilities in practice. The final optimization objective $\mathcal{J}_{\text{Mixed}}$ is a weighted sum of the on-policy and off-policy objectives:

$$\mathcal{J}_{\text{Mixed}}(\theta) = \underbrace{\frac{1}{N_{\text{off}}}(\sum_{j=1}^{N_{\text{off}}} \frac{1}{|o_j|} \sum_{t=1}^{|o_j|} \text{CLIP}(f(\hat{r}_{j,t}(\theta, \phi_j)), \hat{A}^{\text{off}}_{j,t}, \epsilon))}_{\text{off-policy objectives}} + \underbrace{\frac{1}{T_{\text{on}}}(\sum_{i=1}^{N_{\text{on}}} \sum_{t=1}^{|o_i|} \text{CLIP}(r_{i,t}(\theta), \hat{A}^{\text{on}}_{i,t}, \epsilon))}_{\text{on-policy objective}},$$

$$\tag{7}$$

where the clipped surrogate objective $\text{CLIP}(r, A, \epsilon) = min[r \cdot A, \text{clip}(r, 1 - \epsilon, 1 + \epsilon) \cdot A]$, the importance sampling ratios are $\hat{r}_{j,t}(\theta, \phi_j) = \frac{\pi_\theta(o_{j,t}|q, o_{j,<t})}{\pi_{\phi_j}(o_{j,t}|q, o_{j,<t})}$ for off-policy and $r_{i,t}(\theta) = \frac{\pi_\theta(o_{i,t}|q, o_{i,<t})}{\pi_{\theta_{\text{old}}}(o_{i,t}|q, o_{i,<t})}$ for on-policy. The total on-policy token count is $T_{\text{on}} = \sum_{i=1}^{N_{\text{on}}} |o_i|$ and shaping function $f(x) = \frac{x}{x+0.1}$, adapted from Yan et al. (2025) to encourage the model to learn from high-quality off-policy traces that the current policy might assign low probability, preventing the model from merely imitating simple tokens it has already mastered.

As shown in Equation 7, we employ **sequence-level** loss aggregation for the off-policy objectives and token-level aggregation for the on-policy objective Yu et al. (2025a). This design stems from the fact that different off-policy responses may originate from distinct teacher models. If all tokens were aggregated with the same weight, sequences from longer teacher responses would disproportionately influence the gradient, introducing learning biases. To ensure each off-policy response is given equal weighting, regardless of its length or origin, we adopt sequence-level aggregation. When no off-policy responses are present ($N_{\text{off}} = 0$), our objective seamlessly reverts to the GRPO.

## 4 EXPERIMENTS

### 4.1 EXPERIMENTAL SETUP

**Models and Data.** We employed AceReason-Nemotron-1.1-7B Liu et al. (2025d), DeepSeek-R1-Distill-Qwen-7B Guo et al. (2025), OpenR1-Qwen-7B Hugging Face (2025), and Qwen3-8B Team (2025) (thinking mode) as LongCoT teacher models. Experiments primarily employed Qwen2.5-7B-Ins Team (2024) as the base model, extending Qwen2.5-1.5B-Ins Team (2024) and LLaMA3.2-8B-Ins Dubey et al. (2024) to validate AMPO's effectiveness. Based on these teachers, we constructed a novel high-quality dataset of **8.5k samples** by curating verified solutions for problems sourced from the public OpenR1-Math-46k-8192 dataset [1] Yan et al. (2025). This multi-teacher curation process is central to our study. For comprehensive descriptions of the teacher models and the detailed dataset construction methodology, please refer to Appendix A.2.1 and A.2.2. All training hyperparameters are provided in Appendix A.2.4.

**Evaluation and Baselines.** We evaluated our method on six mathematical reasoning benchmarks: AIME2024, AIME2025, AMC Li et al. (2024), Minerva Lewkowycz et al. (2022), Olympiad-Bench He et al. (2024), and Math500 Hendrycks et al. (2021) and four out-of-distribution (OOD) benchmarks: ARC-c Clark et al. (2018), GPQA-diamond (GPQA*) Rein et al. (2024), MMLU-Pro Wang et al. (2024) and HumanEval Chen (2021) to assess generalization. For AIME2024, AIME2025, and AMC, we report Avg@32; for other benchmarks, we report Pass@1. We employed a temperature of 0.6 for evaluation. We compare AMPO against the following baselines: 1) **SFT:** Fine-tuning with SFT using the 32k dataset constructed from all four teacher models (see Appendix A.2.2), effectively serving as a multi-teacher distillation baseline; 2) **GRPO** Shao et al. (2024): Fine-tuning with GRPO on our 8.5k dataset; 3) **SFT+GRPO**: The SFT-tuned model, further trained with the GRPO; 4) **LUFFY**[2] Yan et al. (2025): A Mixed-Policy RL method with OpenR1-Math-46k-8192 dataset representing the single-teacher paradigm. 5) **AMPO:** Our primary method using a Multi-Guidance Pool composed of four LongCoT teacher models; 6)**AMPO (R1 only):** A variant using only the single strong teacher (DeepSeek-R1 Guo et al. (2025)).

### 4.2 MAIN RESULTS

Table 1: **In-Distribution Performance**: Comparison of mathematical reasoning capabilities across different methods.

| Method | AIME24/25 | AMC | MATH500 | Minerva | Olympiad | Avg. |
|---|---|---|---|---|---|---|
| Qwen2.5-1.5B-Ins | 2.8/1.3 | 21.9 | 51.4 | 19.1 | 19.1 | 19.3 |
| SFT | 0.8/0.2 | 13.6 | 35.6 | 7.0 | 12.0 | 11.8 |
| GRPO | 2.9/0.8 | 24.0 | 53.8 | 17.0 | 19.1 | 19.6 |
| SFT+GRPO | 1.8/0.3 | 24.5 | 52.8 | 11.0 | 19.3 | 18.4 |
| **AMPO** | 4.9/1.9 | 28.5 | 58.2 | 17.6 | 23.9 | **22.5** |
| Llama3.2-8B-Ins | 2.8/0.4 | 20.2 | 45.4 | 19.1 | 14.5 | 17.1 |
| SFT | 1.8/0.8 | 18.6 | 46.6 | 20.6 | 17.9 | 17.7 |
| GRPO | 2.1/0.1 | 16.7 | 44.2 | 24.6 | 12.0 | 16.6 |
| SFT+GRPO | 4.2/0.2 | 19.4 | 51.0 | 22.4 | 20.1 | 19.6 |
| **AMPO** | 9.4/1.7 | 26.9 | 59.6 | 37.9 | 25.7 | **24.5** |
| Qwen2.5-7B-Ins | 12.3/6.5 | 43.8 | 75.8 | 36.4 | 38.8 | 35.6 |
| SFT | 12.7/14 | 41.0 | 76.0 | 30.5 | 36.4 | 35.1 |
| GRPO | 11.3/9.9 | 46.6 | 76.8 | 34.6 | 37.6 | 36.1 |
| SFT+GRPO | 20.1/16.7 | 53.4 | 81.0 | 35.0 | 50.0 | 42.7 |
| LUFFY | 16.4/15.2 | 49.3 | 81.6 | 37.9 | 45.8 | 41.0 |
| **AMPO** | 14.0/14.1 | 48.0 | 80.8 | 39.3 | 45.9 | 40.4 |
| AMPO(R1 only) | 14.8/9.2 | 48.8 | 77.0 | 35.7 | 42.5 | 38.0 |
| **SFT+AMPO** | 19.0/18.3 | 53.2 | 82.0 | 37.5 | 49.5 | **43.2** |

[1]https://huggingface.co/datasets/Elliott/Openr1-Math-46k-8192

[2]We download from https://huggingface.co/Elliott/LUFFY-Qwen-Instruct-7B which using Qwen2.5-7B-Ins as base model with more data(46k) from DeepSeek-R1 and re-evaluate it.

**Reasoning Performance.** Our primary results are presented in Table 1. For Qwen2.5-7B-Ins, AMPO demonstrates clear superiority over foundational methods across six mathematical reasoning benchmarks, surpassing SFT and GRPO by **5.3%** and **4.3%** in average scores, respectively. Notably, under the same 8.5k data budget, AMPO outperforms the single-teacher AMPO (R1 only) baseline (**40.4%** vs. **38.0%**), empirically validating that diverse peer guidance yields superior learning signals than a single expert. Furthermore, AMPO matches the performance of LUFFY (41.0%), a strong single-teacher method trained on 5× more data (46k samples), highlighting the data efficiency of our multi-guidance approach.

This advantage is even more pronounced on OOD benchmarks (Table 2), where AMPO's average score of **68.5%** substantially outperforms all baselines, including GRPO (by **10.9%**) and LUFFY (by **5.5%**). This robust performance, especially on challenging datasets like **GPQA\***, confirms that AMPO's adaptive use of diverse guidance effectively enhances the model's reasoning abilities. Additionally, AMPO maintains robust coding capabilities (81.1% on HumanEval), indicating that complex reasoning gains do not come at the expense of general skills.

Table 2: **Out-of-Distribution Performance**: Generalization capabilities across common-sense reasoning, expert-level science, diverse domain knowledge, and code generation tasks.

| Method | ARC-c | GPQA* | MMLU-Pro | HumanEval | Avg. |
|---|---|---|---|---|---|
| Qwen2.5-1.5B-Ins | 42.3 | 0.5 | 25.1 | 42.1 | 27.5 |
| SFT | 7.9 | 8.1 | 16.5 | 49.4 | 20.5 |
| GRPO | 68.7 | 21.7 | 34.1 | 48.8 | 43.3 |
| SFT+GRPO | 71.8 | 24.7 | 31.7 | 51.2 | 44.9 |
| **AMPO** | 73.5 | 27.8 | 34.1 | 50.0 | **46.3** |
| Llama3.2-8B-Ins | 44.3 | 0.0 | 40.2 | 59.8 | 36.1 |
| SFT | 83.2 | 20.2 | 39.6 | 56.7 | 50.0 |
| GRPO | 78.6 | 9.3 | 38.3 | 57.3 | 45.9 |
| SFT+GRPO | 86.1 | 31.3 | 47.0 | 63.4 | 57.0 |
| **AMPO** | 88.0 | 29.8 | 52.8 | 68.9 | **59.9** |
| Qwen2.5-7B-Ins | 85.1 | 6.6 | 55.8 | 80.5 | 57.0 |
| SFT | 80.0 | 17.2 | 45.2 | 83.0 | 56.3 |
| GRPO | 92.1 | 5.1 | 58.7 | 74.4 | 57.6 |
| SFT+GRPO | 91.0 | 38.4 | 58.7 | 83.5 | 67.9 |
| LUFFY | 92.0 | 21.7 | 60.3 | 78.1 | 63.0 |
| **AMPO** | 92.7 | 40.4 | 59.6 | 81.1 | **68.5** |
| AMPO(R1 only) | 92.9 | 0.5 | 59.7 | 81.1 | 58.6 |
| **SFT+AMPO** | 90.7 | 38.4 | 56.7 | 82.3 | 67.0 |

A noteworthy finding is the exceptional performance of the SFT+GRPO baseline. This model, which was first fine-tuned on data from our multi-teacher pool, attained the top average score among all methods on in-distribution tasks. Its performance significantly exceeds that of baselines in prior studies Yan et al. (2025); Zhang et al. (2025) which also employed an SFT-then-RL pipeline but relied on demonstrations from a single teacher (*e.g.*, DeepSeek-R1). This result strongly validates a core premise of our work: leveraging demonstrations from multiple diverse teachers is inherently superior to relying on a single source, a principle with significant practical implications.

Inspired by this result, we further tested SFT+AMPO to investigate if our method could unlock additional potential. The results confirm this hypothesis. Even when starting from the same powerful checkpoint, AMPO's adaptive multi-guidance mechanism extracts additional learning signals during the RL phase, resulting in a further 0.5% performance improvement over SFT+GRPO. This suggests that AMPO excels not only as a standalone training paradigm but also as a powerful enhancement stage, effectively refining models that have already been exposed to high-quality, diverse data. However, this gain sacrifices generalization: SFT+AMPO underperforms standalone AMPO on OOD benchmarks. This suggests the initial SFT phase, while effective for mastering the training data (*i.e.*, knowledge injection), may slightly constrain the model's adaptability to unseen domains, presenting a new trade-off.

**Extending AMPO to Other Models.** To assess generalizability, we applied AMPO to a smaller model (Qwen2.5-1.5B-Ins) and a different family (Llama3.2-8B-Ins), with results in Table 1 and Table 2. On the smaller Qwen model, AMPO outperformed GRPO by **2.9%** on in-distribution and

**3.0%** on OOD tasks, and significantly surpassed the SFT baseline. The improvements on Llama3.2-8B-Ins were even more striking: AMPO achieved a **7.9%** in-distribution gain over GRPO (which itself degraded performance), and a massive **14.0%** OOD improvement. Most notably, AMPO unlocked a new capability on GPQA*, raising the score from **0.0%** (Base) and **9.3%** (GRPO) to a competitive **29.8%**, demonstrating our framework's ability to expand a model's knowledge boundaries.

**Reasoning Efficiency.** Beyond performance metrics, we also calculated the average response length(Avg.Length) across in-distribution reasoning benchmarks for each method to evaluate their efficiency. Table 3 demonstrates that AMPO exhibits remarkably efficient reasoning, generating shorter solutions(Len: 1268) than SFT(Len: 5962), SFT+GRPO(Len: 3318), and LUFFY(Len: 2228) while maintaining comparable performance. This indicates that AMPO's adaptive replacement mechanism not only enables the model to learn from external guidance for more accurate answers but also conserves computational resources by producing more streamlined reasoning chains.

Table 3: Statistics of the average response length for different methods on the in-distribution dataset based on Qwen2.5-7B-Ins.

| Method | Avg.Length |
|---|---|
| Qwen2.5-7B-Ins | 844 |
| SFT | 5962 |
| GRPO | 881 |
| SFT+GRPO | 3318 |
| LUFFY | 2228 |
| AMPO(ours) | 1268 |
| SFT+AMPO | 4341 |

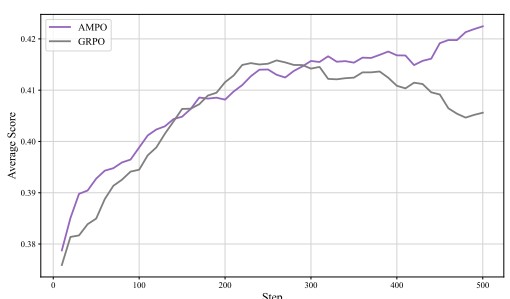

Figure 2: Training Average Score of Validation during GRPO and AMPO training.

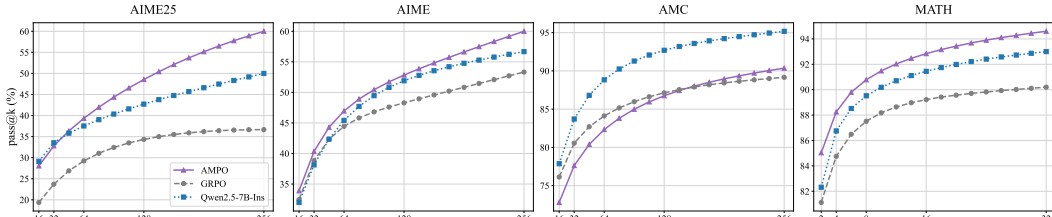

Figure 3: Pass@K Performance with different RL algorithms across several reasoning benchmarks.

**Maintaining Exploration in Reasoning.** To validate AMPO's capability for enhanced exploration, we measured pass@k metrics where higher values indicate a greater capacity to generate diverse solutions Chen et al. (2025); Cheng et al. (2025). We evaluated four reasoning benchmarks, for AIME24/25 and AMC with limited data, we tested the pass@256; for Math500, we assessed the pass@32. Figure 3 demonstrates that AMPO significantly enhances the model's exploration capabilities. On challenging datasets such as AIME24/25, the pass@256 curve markedly outperforms both GRPO and the base model. On the moderately difficult AMC, it also surpasses GRPO. On the relatively straightforward Math500, it similarly outperforms both the base model and GRPO. However GRPO's pass@256 curve consistently lags behind the base model. This indicates that AMPO, through external guidance, effectively introduces richer and more novel knowledge to the model, expanding its capability boundaries while preserving its exploration capacity. It thus strikes an excellent balance between exploration and exploitation.

### 4.3 TRAINING DYNAMICS ANALYSIS

**Training Dynamics of AMPO.** Figure 4a shows that both GRPO and AMPO exhibit an upward trend in reward, indicating gradual performance improvement. Notably, AMPO's reward curve significantly exceeds that of GRPO, demonstrating a more effective and consistent learning signal

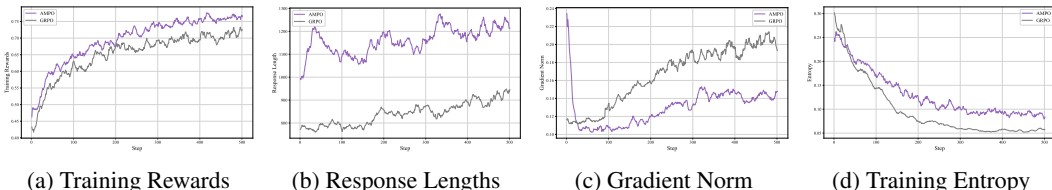

| (a) Training Rewards | (b) Response Lengths | (c) Gradient Norm | (d) Training Entropy |

Figure 4: Training Dynamic of rewards, response lengths, gradient norm and the training entropy during GRPO and AMPO training.

throughout training. Figure 4b indicates that GRPO tends to generate shorter responses during training, potentially leading to insufficient deliberation when encountering challenging tasks. In contrast, AMPO with external guidance, enables the model to engage in more thorough reasoning, producing longer responses while maintaining reasonable inference efficiency without excessively lengthy outputs. Figure 4c illustrates GRPO's training instability during the later stages, while the initial gradient instability in AMPO reflects the challenging but necessary process of assimilating diverse, external knowledge. Crucially, the policy quickly adapts, leading to a more stable training phase compared to the late-stage collapse observed in GRPO.

**Maintaining Exploration in Training.** While Pass@k metrics (Section 4.2) confirm AMPO's superior exploratory outcomes, the training dynamics provide insight into the underlying mechanism. Figure 4d reveals a key difference: while GRPO's policy entropy steadily declines, indicating a collapse into a narrow set of solutions, AMPO maintains a significantly higher entropy throughout training. This sustained exploration, fueled by diverse external guidance, prevents the model from settling into local optima. The direct consequence of this is shown in Figure 2: AMPO's validation score continues to improve across all 500 steps, whereas GRPO's performance stagnates and even degrades in the later stages. This starkly illustrates that AMPO's ability to preserve exploration is critical for achieving robust, long-term performance gains.

### 4.4 ABLATION STUDY AND DISCUSSIONS

Table 4: Ablation Experiments of AMPO.

| Method | AIME24/25 | AMC | MATH500 | Minerva | Olympiad | Avg. | |
| --- | --- | --- | --- | --- | --- | --- | --- |
| | | | | | | $Acc$ | $Len$ |
| **AMPO** | 14.0/14.1 | 48.0 | 80.8 | 39.3 | 45.9 | **40.4** | 1268 |
| -w/o. adaptive | 16.6/12.9 | 49.3 | 82.6 | 34.9 | 46.2 | 40.4 | 2315 |
| -w/o. seq | 15.9/10.6 | 48.7 | 80.4 | 36.0 | 43.7 | 39.2 | 1689 |
| $-r_{length}$ | 14.1/13.1 | 49.6 | 80.2 | 34.6 | 41.9 | 38.9 | 1368 |
| $-r_{random}$ | 14.7/14.9 | 47.0 | 79.8 | 32.4 | 43.4 | 38.7 | **1103** |

**Ablation Study.** We conducted ablation experiments to validate the contribution of each key component in AMPO. The main settings are: 1) **-w/o. adaptive:** Removes the adaptive trigger, always replacing incorrect on-policy responses with guidance; 2) **-w/o. seq:** Replaces sequence-level with token-level aggregation for the off-policy loss; 3) $-r_{length}$: Replaces comprehension-based selection with a heuristic that prioritizes shorter responses; 4) $-r_{random}$: Replaces comprehension-based selection with random selection.

As shown in Table 4, each component is crucial for optimal performance. Removing the adaptive replacement mechanism results in a nearly 2x increase in response length with no corresponding gain in average accuracy, confirming its critical role in maintaining reasoning efficiency. Disabling sequence-level aggregation (-w/o. seq) and replacing our selection mechanism with a length-based heuristic ($-r_{length}$) both lead to significant performance drops of **-1.2%** and **-1.5%**, respectively. Randomly selecting guidance ($-r_{random}$) is the most detrimental, causing a **-1.7%** drop in accuracy, which underscores the importance of the comprehension-based selection strategy.

**Discussion of Guidance Utilization** ($k_0$)**.** We analyzed the impact of the guidance replacement target ($k_0$) with results shown in Table 5 and Figure 5. A clear trend emerges: increasing a fixed

$k_0$ from 1 to 4 generally improves accuracy (peaking at 41.1%) and sustains exploration entropy. This benefit stems from our **Adaptive Multi-Guidance Replacement** strategy, which injects diverse external knowledge *only* during sparse reward episodes, enabling the model to concentrate on learning diverse solutions specifically when facing challenging problems. However, this comes at a cost: higher fixed $k_0$ values lead to significantly longer response lengths and training instability (*e.g.*, gradient spikes at $k_0 = 4$).

Hypothesizing that a dynamic approach could resolve this trade-off, we introduced a **Dynamic $k_0$ Balancing Strategy**. Unlike the fixed replacement triggered only by complete failure, this strategy aims to maintain a balanced ratio of correct-to-incorrect responses (target: 50%) thereby ensuring a contrastive learning signal. Specifically, we modified the original replacement trigger condition to utilize guidance whenever the model's correct response ratio drops below 50% ($N_{\text{correct}} < G/2$), and dynamically set $k = \min(N_g, G/2 - N_{\text{correct}})$.

As shown in Table 6, this dynamic strategy achieves the **highest average accuracy (41.6%)** while maintaining a remarkably concise average length (Len: 1234), comparable to the efficient $k_0 = 1$ setting. This indicates that dynamically determining the necessary guidance is superior to fixed settings. We hope this preliminary finding inspires further research into adaptive teacher interaction.

**Discussion of Teacher Composition.** We further experimented with more different compositions for the Multi-Guidance Pool to understand the impact of teacher choice. Primarily comparing the following configurations: 1) **R1 only:** Using only DeepSeek-R1 Guo et al. (2025); 2) **4 Long-CoTs:** The four LongCoT models mentioned in 4.1 we routinely employed in main experiments; 3) **4 ShortCoTs:** Employing four short chain-of-thought (ShortCoT) reasoning models: Qwen3-8B Team (2025) (no thinking mode), Qwen2.5-Math-7B-Ins Yang et al. (2024), Qwen2.5-Math-7B-Oat-Zero Liu et al. (2025c), and SynLogic-7B Liu et al. (2025a); 4) **4 LongCoTs + R1:** Integrating the four LongCoT models with DeepSeek-R1; 5) **4 LongCoTs + 4 ShortCoTs:** Combining the four LongCoT and four ShortCoT models. Except for 1), where $k_0$ is set to 1, $k_0$ is set to 2 for all other configurations.

The results (Table 7) consistently demonstrate the value of our multi-teacher strategy, as all configurations outperform the GRPO baseline (Table 1). Our main setup, 4 LongCoTs, achieves the highest accuracy (40.4%). Interestingly, a pool of 4 ShortCoTs yields the most efficient responses (942 Avg.Length) but at the cost of accuracy. Therefore, mixing LongCoT models with ShortCoT models can reduce the reasoning length while ensuring performance, achieving more efficient reasoning. Notably, the single-teacher baseline using the powerful DeepSeek-R1 model is defeated by most multi-teacher configurations, even those using smaller models, *e.g.*, 7B models. This provides strong evidence for our core hypothesis: a set of diverse teachers provides a more robust and effective learning signal than a single, powerful expert. While our experiments confirm the benefits of teacher diversity, we note that a deeper exploration of optimal teacher combinations is a promising direction for future research.

## 5 CONCLUSION

In this work, we identify and address a key limitation in existing Mixed-Policy RL methods: their reliance on a single, powerful off-policy teacher, which tends to constrain exploratory diversity. Drawing inspiration from multi-teacher strategies in knowledge distillation, we propose **AMPO**, a novel framework that intelligently leverages guidance from multiple, diverse teacher models, intervening only when the on-policy model fails. Our two core contributions, **Adaptive Multi-Guidance Replacement** and **Comprehension-based Guidance Selection**, ensure that this external knowledge is used both efficiently and effectively. Extensive experiments demonstrate that AMPO significantly outperforms its GRPO baseline by **4.3%** on mathematical reasoning tasks and a striking **10.9%** on out-of-distribution tasks, showcasing superior performance and generalization.

ETHICS STATEMENT

Our work focuses on enhancing the mathematical reasoning capabilities of Large Language Models, a technical domain with no direct engagement with human subjects or personally identifiable data. The training dataset was derived from a publicly available resource, OpenR1-Math-46k-8192 Yan et al. (2025), and curated through a verifiable process detailed in Appendix A.2.2. We acknowledge the general dual-use concerns associated with advancing powerful AI models, however, our research does not introduce novel applications that pose foreseeable societal harm. The use of AI as an assistive tool in the preparation of this manuscript is disclosed in Appendix A.4.

REPRODUCIBILITY STATEMENT

To ensure full reproducibility, our source code is available at `https://anonymous.4open.science/r/7fBQd46C`. We also commit to releasing our code, curated training dataset and final model weights. Our experimental framework is detailed throughout the paper for transparency. The composition of the Multi-Guidance Pool and the models used are described in Appendix A.2.1. The dataset construction process is detailed in Appendix A.2.2, and all training hyperparameters for RL and SFT setups are specified in Appendix A.2.4. A detailed analysis of the required computational resources is provided in Appendix A.3.3. All evaluation benchmarks are publicly available and listed in Section 4.1.

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

# A APPENDIX

## A.1 LIMITATIONS AND FUTURE WORK

While our work provides a foundational exploration of applying multi-teacher strategies to Mixed-Policy RL, we acknowledge several limitations that offer promising avenues for future research. First, the construction of our Multi-Guidance Pool relied on heuristic selection, future work could explore systematic methods for composing teacher pools and developing **theoretical metrics to quantify reasoning diversity** (*e.g.*, trajectory distance). Second, regarding **scalability**, while we utilized up to 8 teachers, our framework is designed to scale to larger pools (*e.g.*, 10+) as the selection cost increases linearly; validating this at scale is a future priority. Third, our framework currently utilizes only a single reasoning path from each teacher; incorporating multiple diverse responses could provide richer signals. Additionally, our analysis of key hyperparameters ($k_0, G$) warrants more exhaustive investigation. Lastly, we aim to apply AMPO to more diverse domains beyond mathematics to create a more generalized reasoning agent.

## A.2 EXPERIMENT DETAILS

### A.2.1 COMPOSITION OF MULTI-GUIDANCE POOL

**Main Experiments Teachers (4 LongCoTs).** The four primary teacher models were selected for their strong long chain-of-thought (LongCoT) reasoning capabilities:

- **AceReason-Nemotron-1.1-7B** Liu et al. (2025d): A 7B parameter model from NVIDIA, developed by applying a combination of Supervised Fine-Tuning (SFT) and multi-stage Reinforcement Learning (RL) on Qwen2.5-Math-7B.
- **DeepSeek-R1-Distill-Qwen-7B** Guo et al. (2025): A 7B parameter model created by DeepSeek-AI through knowledge distillation. It uses the powerful DeepSeek-R1 as the teacher model to fine-tune a Qwen2.5-Math-7B student model, transferring its advanced reasoning capabilities.
- **OpenR1-Qwen-7B** Hugging Face (2025): A 7B parameter fully open reproduction of early DeepSeek-R1 capabilities, developed by fine-tuning a Qwen2.5-Math-7B-Instruct base model on the extensive OpenR1-220k-Math dataset.
- **Qwen3-8B** Team (2025): A 8B parameter powerful base model from Alibaba Cloud's Qwen3 series. It is designed for balanced performance in both complex reasoning and general conversational tasks. We utilize its "thinking mode" for main experiments.

**Discussion Experiments Teachers ("R1 only", "4 ShortCoTs").** For the ablation study on teacher composition, we included additional models representing different capabilities and reasoning styles:

- **DeepSeek-R1** Guo et al. (2025): The 671B parameter powerful, proprietary reasoning model from DeepSeek-AI used as a single-teacher baseline. It was developed through a sophisticated multi-stage pipeline involving both RL and SFT.
- **Qwen2.5-Math-7B-Ins** Yang et al. (2024): An 7B parameter instruction-tuned model from Alibaba Cloud, specifically optimized for mathematical reasoning tasks using both Chain-of-Thought and Tool-integrated Reasoning data.
- **Qwen2.5-Math-7B-Oat-Zero** Liu et al. (2025c): A 7B parameter model derived from Qwen2.5-Math-7B through a pure reinforcement learning process (similar to R1-Zero), which aims to unlock reasoning capabilities without relying on expert demonstrations.
- **SynLogic-7B** Liu et al. (2025a): A 7B parameter model built on the Qwen2.5-7B-Base model and enhanced with reinforcement learning (RL) using the specialized SynLogic dataset fine-tuned specifically on the SynLogic dataset, specifically for advanced logical reasoning.
- **Qwen3-8B** Team (2025): A 8B parameter powerful base model from Alibaba Cloud's Qwen3 series. It is designed for balanced performance in both complex reasoning and general conversational tasks. We utilize its "no thinking mode" for discussion experiments.

### A.2.2 DATASET CONSTRUCTION DETAILS

To ensure a fair comparison across all experimental settings, particularly the analysis of different teacher compositions (the results of which are in Table 7), our datasets were constructed through

a rigorous, multi-stage process designed to maintain a consistent set of questions for all models. ==Constructing this multi-teacher dataset was computationally intensive. We utilized **4 NVIDIA RTX 5090 GPUs** running continuously for approximately **one week** to collect reasoning paths from **4 LongCoT and 4 ShortCoT models**. The extended duration is primarily attributed to the lower inference efficiency associated with LongCoT models. This timeframe encompasses the generation of diverse reasoning paths and the subsequent verification steps.==

1. **Source Data:** We began with the 10,000 question-answer (QA) pairs from the OpenR1-Math-46k-8192 dataset Yan et al. (2025).

2. **Per-Composition Data Curation:** For each teacher composition group defined in our study, we curated a corresponding set of demonstrations.
   - For compositions involving our selected LongCoT or ShortCoT models, we executed a full generation pipeline:
   (a) **Generation:** Each teacher model within the group was prompted to generate 8 distinct reasoning paths for every question.
   (b) **Verification:** All generated paths were evaluated for correctness using Math-Verify [3], and incorrect responses were discarded.
   (c) **Selection:** For each question, if a teacher produced multiple correct paths, we retained only its shortest correct path to ensure conciseness.
   - For the baseline "R1 only" composition, we directly used the high-quality demonstrations already provided in the source dataset, as they were originally generated by DeepSeek-R1.

3. **Dataset Unification via Intersection:** After curating data for every teacher composition, we obtained multiple candidate datasets. To create a unified set of questions for all our experiments, we took the **intersection of the question sets** from all these candidate datasets. This crucial step ensures that for every question in our final dataset, there is at least one correct demonstration available from every teacher composition group we tested.

4. **Final RL Dataset Curation:** The intersection process yielded our final RL dataset of **8,491** unique QA pairs. The **Multi-Guidance Pool** ($\mathcal{P}_G$) for each question is therefore dynamic: for any given experiment (*e.g.*, training with the "4 LongCoTs" composition), the pool for a question contains the correct demonstrations **only from those teachers in the group who successfully solved it**. Thus, the number of demonstrations per question can range from one to the total number of teachers in the composition.

The **SFT dataset**, used for our main baseline comparison, was then created from the complete set of demonstrations generated by our primary "4 LongCoTs" teacher composition. By treating each of the 8,491 questions and its corresponding correct teacher demonstrations as individual training instances, we expanded the data into a comprehensive SFT dataset of **32,252** pairs.

### A.2.3 System Prompt and Reward Function Design

To ensure consistent and automated evaluation for both our reward function and comprehension score ($r_p$), we used a standardized concise system prompt for all models. The prompt instructs the model to provide a step-by-step reasoning process and enclose the final answer in <answer></answer>tags consistent with the Yu et al. (2025b):

---

**System Prompt**

You are a helpful assistant. Please reason step by step to solve the problem and put the final answer within the <answer></answer>tags.

**User:** {QUESTION}
**Assistant:**

---

To encourage the model to place answers within the designated labels, we set the following composite reward function:

$$R(\cdot) = (1 - \beta) \cdot R_{accuracy}(\cdot) + \beta \cdot R_{format} \tag{8}$$

---
[3]https://github.com/huggingface/Math-Verify

Here, $R_{accuracy}$ is a binary reward (1 if the final answer is correct, 0 otherwise) from a rule-based verifier. Similarly, $R_{format}$ is a binary reward (1 if the specified tags are correctly used, 0 otherwise). We set the weighting coefficient $\beta = 0.1$.

Therefore, we set the sparse reward threshold $\tau = 0.5$ (as mentioned in Section 3.2). This configuration ensures that adaptive replacement is triggered if and only if every on-policy response in a batch is factually incorrect, regardless of its formatting.

### A.2.4 TRAINING HYPERPARAMETERS

**RL Training.** All reinforcement learning experiments were conducted using the VERL Sheng et al. (2024) framework on 4 NVIDIA H100 GPUs, except for the smaller Qwen2.5-1.5B-Ins which used a single NVIDIA H100 GPU. Models were trained for 500 steps using training batch size of 128 and mini-batch size of 64 for updates. For each prompt during rollout, we sampled $G = 8$ responses with a temperature of 1.0 and a maximum response length of 8192 tokens. The optimization used a learning rate of 1e-6, a clipping ratio of 0.2, and an entropy coefficient of 0.001 to encourage exploration. For our AMPO method, the default number of guidance replacements was set to $k_0 = 2$.

**SFT Training.** The supervised fine-tuning (SFT) baseline was trained using the LLaMA-Factory Zheng et al. (2024) library on a single NVIDIA L40 GPU. We employed the LoRA (Low-Rank Adaptation) Hu et al. (2022) technique, applying it to all linear modules of the model. The model was trained for 3 epochs with a learning rate of 5e-5, which was controlled by a cosine learning rate scheduler with a warmup ratio of 0.1. We used batch size of 1 and gradient accumulation steps of 8. Our SFT training was performed using bf16 precision, and the maximum sequence length was capped at 8192 tokens.

## A.3 OTHER EXPERIMENTS RESULTS

### A.3.1 DISCUSSION OF GUIDANCE UTILIZATION ($k_0$)

Table 5: The Performance of different $k_0$-value with AMPO.

| $k_0$ | AIME24/25 | AMC | MATH500 | Minerva | Olympiad | Avg. |
|---|---|---|---|---|---|---|
| 1 | 16.0/11.4 | 49.4 | 80.4 | 36.0 | 43.1 | 39.4 |
| 2 | 14.0/14.1 | 48.0 | 80.8 | 39.3 | 45.9 | 40.4 |
| 3 | 17.3/11.5 | 50.0 | 81.0 | 38.2 | 44.3 | 40.4 |
| 4 | 15.1/14.1 | 53.0 | 79.8 | 37.5 | 47.0 | **41.1** |

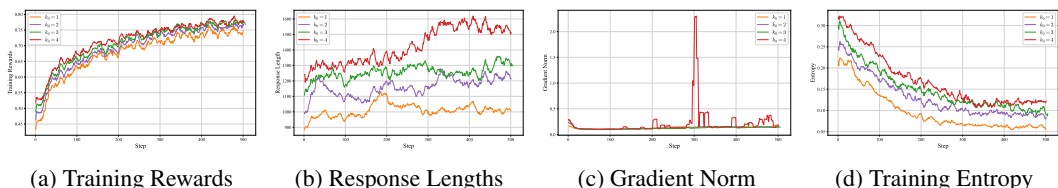

(a) Training Rewards     (b) Response Lengths     (c) Gradient Norm     (d) Training Entropy

Figure 5: Training Dynamic of rewards, response lengths, gradient norm and the training entropy during AMPO training with different $k_0$.

Table 6: Impact of Fixed vs. Dynamic $k_0$ (Adaptive) on performance and efficiency.

| Setting | AIME24/25 | AMC | MATH500 | Minerva | Olympiad | Avg. Acc | Avg. Len |
|---|---|---|---|---|---|---|---|
| Fixed $k_0 = 1$ | 16.3/11.0 | 50.2 | 78.2 | 36.8 | 40.6 | 38.8 | **1104** |
| Fixed $k_0 = 2$ | 17.5/12.2 | 52.6 | 80.6 | 37.1 | 46.1 | 41.0 | 2439 |
| Fixed $k_0 = 3$ | 16.9/12.5 | 51.2 | 81.2 | 37.9 | 47.0 | 41.1 | 2103 |
| Fixed $k_0 = 4$ | 16.6/15.3 | 51.3 | 82.2 | 38.2 | 45.3 | 41.5 | 1750 |
| Dynamic | 17.4/17.7 | 50.5 | 80.2 | 39.0 | 44.7 | **41.6** | 1234 |

### A.3.2 DISCUSSION OF TEACHER COMPOSITION

Table 7: The Performance of different composition of Multi-Guidance Pool with AMPO.

| Composition of Teachers | AIME24/25 | AMC | MATH500 | Minerva | Olympiad | Avg. Acc | Avg. Len |
|---|---|---|---|---|---|---|---|
| R1 only | 14.8/9.2 | 48.8 | 77.0 | 35.7 | 42.5 | 38.0 | 1430 |
| 4 LongCoTs | 14.0/14.1 | 48.0 | 80.8 | 39.3 | 45.9 | **40.4** | 1268 |
| 4 ShortCoTs | 16.1/11.3 | 46.7 | 79.2 | 34.2 | 39.4 | 37.8 | **942** |
| 4 LongCoTs + R1 | 18.1/10.1 | 50.2 | 80.0 | 37.1 | 43.4 | 39.8 | 1378 |
| 4 LongCoTs + 4 ShortCoTs | 17.0/11.7 | 51.5 | 79.2 | 35.3 | 42.3 | 39.6 | 1205 |

### A.3.3 COMPUTATIONAL RESOURCE ANALYSIS

Table 8: Comparison of resource requirements for GRPO, AMPO with ablation setup.

| Method | GPU Hours |
|---|---|
| GRPO | 22x4 |
| AMPO | 32x4 |
| AMPO -w/o adaptive | 47x4 |
| AMPO -w/o seq | 36x4 |
| AMPO -r_length | 29x4 |
| AMPO -r_random | 27x4 |

Table 9: Comparison of resource requirements for GRPO, AMPO with different composition of Multi-Guidance-Pool.

| Method | GPU Hours |
|---|---|
| GRPO | 22x4 |
| AMPO (R1 only) | 32x4 |
| AMPO (4 LongCoTs) | 32x4 |
| AMPO (4 ShortCoTs) | 23x4 |
| AMPO (4 LongCoTs + R1) | 33x4 |
| AMPO (4 LongCoTs + 4ShortCoTs) | 31x4 |

As detailed in Table 8, the standard AMPO configuration incurs a moderate computational overhead (32x4 GPU hours) compared to the GRPO baseline (22x4). This increase is justified by the significant performance gains observed. The efficiency of our adaptive replacement strategy is highlighted by the substantial cost increase (47x4 hours) when it is disabled (-w/o adaptive). Our comprehension-based selection mechanism adds only a minor, well-justified overhead compared to simpler heuristics.

Furthermore, Table 9 demonstrates that the choice of teacher composition can effectively manage training costs. Notably, the AMPO (4 ShortCoTs) configuration requires only 23x4 GPU hours, nearly matching the GRPO baseline. This efficiency stems from the shorter response lengths generated by ShortCoT models (as shown in Table 7), which accelerates training. This result confirms that AMPO can achieve superior performance with a computational cost comparable to the baseline, underscoring its practicality.

### A.3.4 CASE STUDIES

We provide a comparison of the reasoning processes of the base model, the GRPO-trained model, and the AMPO-trained model on a challenging mathematical problem. This case studies highlights the distinct differences in their problem-solving approaches.

**GRPO-trained Model.** While the model trained with GRPO shows signs of improved reasoning, such as using reflective phrases like "We need to check...", its approach remains superficial. It correctly identifies the first scenario for a unique solution, where the derived quadratic equation has exactly one root, but fails to explore other possibilities. This limited depth of reasoning leads it to the same incorrect conclusion as the base model.

**AMPO-trained Model.** In stark contrast, the AMPO-trained model demonstrates a significantly more sophisticated and robust reasoning process. It exhibits genuine **meta-cognition and self-correction**, explicitly identifying a potential flaw in a simplistic approach with the phrase, "There seems to be a misunderstanding...Let's re-examine...". Critically, it successfully deconstructs the problem into two distinct cases: (1) the quadratic equation having one valid root, and (2) the quadratic having two roots where only one satisfies the domain constraints of the original logarithmic equation. This comprehensive analysis and deeper deliberation allow it to navigate the problem's complexity and arrive at the correct solution.

---

### Problem Prompt

Find the number of integer values of $k$ in the closed interval $[-500, 500]$ for which the equation $\log(kx) = 2\log(x+2)$ has exactly one real solution. (**Answer: 501**)

---

### Response generated by Qwen2.5-7B-Instruct

To solve the equation $\log(kx) = 2\log(x+2)$ and find the number of integer values of $k$ in the interval $[-500, 500]$ for which the equation has exactly one real solution, we will follow these steps:

**1. Rewrite the equation using properties of logarithms:**

$$\log(kx) = 2\log(x+2)$$

Using the property $\log(a) = b\log(c) \implies \log(a) = \log(c^b)$, we get:

$$\log(kx) = \log((x+2)^2)$$

**2. Equate the arguments of the logarithms:**

$$kx = (x+2)^2$$

**3. Expand and rearrange the equation:**

$$kx = x^2 + 4x + 4$$

$$x^2 + (4-k)x + 4 = 0$$

**4. Analyze the quadratic equation:** The quadratic equation $x^2 + (4-k)x + 4 = 0$ will have real solutions if its discriminant is non-negative. The discriminant $\Delta$ of a quadratic equation $ax^2 + bx + c = 0$ is given by:

$$\Delta = b^2 - 4ac$$

Here, $a = 1$, $b = 4-k$, and $c = 4$. Thus,

$$\Delta = (4-k)^2 - 4 \cdot 1 \cdot 4$$

$$\Delta = (4-k)^2 - 16$$

$$\Delta = k^2 - 8k$$

**5. Determine when the discriminant is non-negative:**

$$k^2 - 8k \geq 0$$

Factor the quadratic expression:

$$k(k-8) \geq 0$$

This inequality holds when $k \leq 0$ or $k \geq 8$.

**6. Ensure the quadratic equation has exactly one real solution:** For the quadratic equation to have exactly one real solution, the discriminant must be zero:

$$k(k - 8) = 0$$

This gives us two solutions:

$$k = 0 \quad \text{or} \quad k = 8$$

**7. Check the conditions for these values of $k$:**
For $k = 0$:

$$x^2 + 4x + 4 = 0$$
$$(x + 2)^2 = 0$$
$$x = -2$$

However, $x = -2$ is not in the domain of the original logarithmic equation because $\log(x + 2)$ is undefined for $x = -2$. Thus, $k = 0$ does not provide a valid solution.
For $k = 8$:

$$x^2 - 4x + 4 = 0$$
$$(x - 2)^2 = 0$$
$$x = 2$$

For $x = 2$:

$$\log(8 \cdot 2) = 2\log(2 + 2)$$
$$\log(16) = 2\log(4)$$
$$\log(16) = \log(16)$$

This is true, and $x = 2$ is valid.
**8. Count the valid integer values of $k$:** The only valid integer value of $k$ in the interval $[-500, 500]$ is $k = 8$.

Therefore, the number of integer values of $k$ for which the equation has exactly one real solution is $\boxed{1}$.

(**response tokens: 933, correctness: False**)

---

Response generated by the model trained with GRPO

To find the number of integer values of $k$ in the interval $[-500, 500]$ for which the equation $\log(kx) = 2\log(x + 2)$ has exactly one real solution, we start by manipulating the logarithmic equation.

Using the properties of logarithms, we can rewrite the equation as:

$$\log(kx) = \log((x + 2)^2)$$

This implies:

$$kx = (x + 2)^2$$

Expanding the right side, we get:

$$kx = x^2 + 4x + 4$$

Rearranging terms gives us a quadratic equation:

$$x^2 + (4 - k)x + 4 = 0$$

For this quadratic equation to have exactly one real solution, its discriminant must be zero. The discriminant $\Delta$ of the quadratic equation $ax^2 + bx + c = 0$ is given by $\Delta = b^2 - 4ac$. Here, $a = 1$, $b = 4 - k$, and $c = 4$, so the discriminant is:

$$\Delta = (4 - k)^2 - 4 \cdot 1 \cdot 4 = (4 - k)^2 - 16$$

Setting the discriminant to zero for exactly one solution, we have:
$$(4 - k)^2 - 16 = 0$$

This can be rewritten as:
$$(4 - k)^2 = 16$$

Taking the square root of both sides, we get:
$$4 - k = 4 \quad \text{or} \quad 4 - k = -4$$

Solving these equations, we find:
$$k = 0 \quad \text{or} \quad k = 8$$

**We need to check** if these values of $k$ indeed give exactly one solution for $x$ and satisfy the domain restrictions of the original logarithmic equation $\log(kx)$ and $\log(x + 2)$, which require $kx > 0$ and $x + 2 > 0$ (i.e., $x > -2$).

1. For $k = 0$, the equation becomes:
$$0 \cdot x = (x + 2)^2 \implies 0 = x^2 + 4x + 4 \implies (x + 2)^2 = 0 \implies x = -2$$

But $x = -2$ is not in the domain $x > -2$, so $k = 0$ is not a valid solution.

2. For $k = 8$, the equation becomes:
$$8x = (x+2)^2 \implies 8x = x^2+4x+4 \implies x^2-4x+4 = 0 \implies (x-2)^2 = 0 \implies x = 2$$

For $x = 2$, **we check the domain conditions**:
$$kx = 8 \cdot 2 = 16 > 0$$
$$x + 2 = 2 + 2 = 4 > 0$$

Both conditions are satisfied, so $k = 8$ is a valid solution.

Next, **we need to check** if there are any other integer values of $k$ that could give exactly one solution.

For the quadratic equation $x^2+(4-k)x+4 = 0$ to have exactly one solution, the discriminant must be zero, which we already used to find $k = 8$.

For the quadratic to have no real solutions, the discriminant must be negative:
$$(4 - k)^2 - 16 < 0 \implies (4 - k)^2 < 16 \implies -4 < 4 - k < 4 \implies 0 < k < 8$$

**We need to check** if any integer $k$ in the interval $0 < k < 8$ can give exactly one solution. If the quadratic has a negative discriminant, it has no real solutions. If the quadratic has a positive discriminant, it has two distinct real solutions. **We need to ensure** there are no other $k$ values that can give exactly one solution besides $k = 8$.

For $k$ in the interval $(-500, 500]$, we need to ensure that $kx = (x+2)^2$ does not yield another solution for integer $k$ other than $k = 8$. The quadratic equation $x^2 + (4 - k)x + 4 = 0$ will have exactly one solution if and only if the discriminant is zero, which we have already shown happens only for $k = 8$.

Therefore, the only integer value of $k$ that satisfies the condition is $k = 8$. Since we are asked for the number of integer values of $k$ in the interval $[-500, 500]$, and we have found only one such value, the answer is: **<answer>1 </answer>**.

(**response tokens: 1145, correctness: False**)

### Response generated by the model trained with AMPO

To solve the equation $\log(kx) = 2\log(x + 2)$ for integer values of $k$ in the interval $[-500, 500]$ such that the equation has exactly one real solution, we start by manipulating the logarithmic equation.

First, we rewrite the equation using properties of logarithms:

$$\log(kx) = 2\log(x + 2)$$

This can be rewritten as:

$$\log(kx) = \log((x + 2)^2)$$

Since the logarithmic function is one-to-one, we can equate the arguments:

$$kx = (x + 2)^2$$

Expanding the right side, we get:

$$kx = x^2 + 4x + 4$$

Rearranging terms to form a quadratic equation in $x$:

$$x^2 + (4 - k)x + 4 = 0$$

For this quadratic equation to have real solutions, the discriminant must be non-negative. The discriminant $\Delta$ is:

$$\Delta = (4 - k)^2 - 4 \cdot 1 \cdot 4 = (4 - k)^2 - 16$$

We need $\Delta \geq 0$:

$$(4 - k)^2 - 16 \geq 0$$

This inequality can be factored as:

$$((4 - k) - 4)((4 - k) + 4) \geq 0 \implies (-k)(8 - k) \geq 0$$

which simplifies to:

$$(k - 8)k \geq 0$$

The solutions to this inequality are $k \leq 0$ or $k \geq 8$. **However, we must also satisfy** the domain conditions of the original equation, primarily $kx > 0$ and $x > -2$. If a solution $x$ is found, these must hold.

The quadratic equation $x^2 + (4 - k)x + 4 = 0$ gives the potential solutions. We need exactly one of its roots to be a valid solution to the original logarithmic equation.

**1. Case 1: The quadratic has one real root (a repeated root).** This occurs when the discriminant is zero:

$$(4 - k)^2 - 16 = 0 \implies (k - 4)^2 = 16 \implies k - 4 = \pm 4 \implies k = 8 \text{ or } k = 0$$

- For $k = 8$, the quadratic is $x^2 - 4x + 4 = 0 \implies (x - 2)^2 = 0$, giving the solution $x = 2$. **We check the domain**: $kx = 8 \cdot 2 = 16 > 0$ and $x + 2 = 4 > 0$. Both are valid. So, $k = 8$ works.

- For $k = 0$, the quadratic is $x^2 + 4x + 4 = 0 \implies (x + 2)^2 = 0$, giving $x = -2$. **We check the domain**: $x + 2 > 0$ is not satisfied. So, $k = 0$ does not work.

**2. Case 2: The quadratic has two distinct real roots.** This occurs when the discriminant is positive, i.e., $k < 0$ or $k > 8$. Let the roots be $x_1$ and $x_2$. From Vieta's formulas, the product of the roots is $x_1 x_2 = 4$. Since the product is positive, the two roots must have the same sign. The sum of the roots is $x_1 + x_2 = -(4 - k) = k - 4$.

- For $k > 8$: The sum $k - 4$ is positive. Since both the sum and product are positive, both roots $x_1$ and $x_2$ must be positive. For each positive root $x$, the domain condition $kx > 0$ is

satisfied (since $k > 8$). The condition $x + 2 > 0$ is also satisfied. This means for any $k > 8$, we get two distinct, valid solutions. **This is not what we want**.

- For $k < 0$: The sum $k - 4$ is negative. Since the sum is negative and the product is positive, both roots $x_1$ and $x_2$ must be negative. **Let's check** the domain for a negative root $x < 0$. The condition $x + 2 > 0$ might hold (if $-2 < x < 0$), but the condition $kx > 0$ requires that if $k < 0$, then $x$ must also be negative, which is true for both roots. So for every integer $k < 0$, we have two negative roots, both of which are valid solutions. **This also gives two solutions, not one**.

**There seems to be a misunderstanding** in the provided text's reasoning. **Let's re-examine** the problem based on the idea that one of the quadratic's roots might be invalid. A root $x$ of $x^2 + (4 - k)x + 4 = 0$ is only a valid solution to the original equation if $x + 2 > 0$ and $kx > 0$. A root is invalid if $x \leq -2$ or if $kx \leq 0$.

Let $f(x) = x^2 + (4 - k)x + 4$. If the quadratic has two roots, $x_1, x_2$, but only one is valid, that gives one solution.

**Let's reconsider** $k < 0$. We established both roots $x_1, x_2$ are negative. Product $x_1 x_2 = 4$. Sum $x_1 + x_2 = k - 4 < -4$.

Since the product is 4, neither root can be 0. Since the sum is less than -4, at least one root must be less than -2.

**Let's check** the value of $f(-2) = (-2)^2 + (4 - k)(-2) + 4 = 4 - 8 + 2k + 4 = 2k$. For $k < 0$, $f(-2) = 2k < 0$. This means $-2$ lies between the two roots $x_1$ and $x_2$.

So, we have $x_1 < -2 < x_2 < 0$. - The root $x_1$ is invalid because $x_1 + 2 < 0$. - The root $x_2$ is in $(-2, 0)$. For this root, $x_2 + 2 > 0$. Also, since $k < 0$ and $x_2 < 0$, the product $kx_2 > 0$. So this root is valid. Therefore, for every integer $k < 0$, there is exactly one valid real solution.

So the valid integer values for $k$ are: 1. $k = 8$ (one value). 2. All integers $k$ such that $k < 0$. In the interval $[-500, 500]$, this corresponds to $k \in \{-500, -499, \ldots, -1\}$. The number of these integers is 500.

The total number of integer values of $k$ is $500 + 1 = 501$. Thus, the number of integer values of $k$ is <**answer**>**501** </**answer**>.

(**response tokens: 1826, correctness: True**)

## A.4 THE USE OF LARGE LANGUAGE MODELS (LLMs)

The conceptual framework and core ideas outlined in this paper represent the authors' original contributions. AI-driven language models were employed solely as auxiliary tools to support specific, well-defined tasks. These tasks included retrieving relevant literature, implementing utility functions, generating code for data visualization, and assisting with language polishing in manuscript writing.

