# OpenReview forum: "More Than One Teacher: Adaptive Multi-Guidance Policy Optimization for Diverse Exploration"
_ICLR.cc/2026/Conference — Submitted to ICLR 2026_

### Official Review · Reviewer_y1Nj · 2025-10-17

**Soundness:** 2
**Presentation:** 2
**Contribution:** 2
**Rating:** 2
**Confidence:** 4

**Summary:**

the paper introduces a new method that guides the learner dynamically, only when the learner cannot produce any correct traces. It also selectively learns what trace is best for the learner.

**Strengths:**

- relevant topic, writing is mostly clear

**Weaknesses:**

- the method has overall a lack of novelty, it seems to be combining findings existing works, also lacks motivation, justification and theoretical analysis
- requires the teacher log probabilities to calculate the importance sampling term, might be too restrictive for certain scenarios
- the method needs several teacher models, but was only compared to GRPO as the baseline for llama-8b and qwen 1.5b, which is unfair.

[1] Ziru Liu, Cheng Gong, Xinyu Fu, Yaofang Liu, Ran Chen, Shoubo Hu, Suiyun Zhang, Rui Liu,
Qingfu Zhang, and Dandan Tu. Ghpo: Adaptive guidance for stable and efficient llm reinforcement learning.
[2]  Rlpr: Extrapolating rlvr to general domains without verifiers

**Questions:**

- what is the justification of using the shaping function, f(x)?

---

> ### Author Response · Authors · 2025-11-23
> **Response to Reviewer y1Nj (1/2)**
>
> We sincerely thank the reviewer for the critical assessment and for acknowledging that the writing is clear. We value your feedback on novelty and technical details. Below, we respectfully clarify the distinctions of our work and provide comprehensive experimental evidence to address your concerns.
>
> **1. Novelty and Motivation (Weakness 1)**
>
> > *The method has overall a lack of novelty, it seems to be combining findings existing works, also lacks motivation, justification and theoretical analysis*
>
> We respectfully emphasize that **AMPO is not merely a combination of existing modules, but a novel framework specifically designed to introduce Multi-Teacher strategies into the Mixed-Policy RL paradigm.**
>
> * **Differentiation from Prior Work:**
>     Most existing Mixed-Policy RL works, such as **LUFFY** ([Yan et al., 2025](https://arxiv.org/abs/2504.14945)) and **ReLIFT** ([Ma et al., 2025](https://arxiv.org/abs/2506.07527)), typically rely on a **single** off-policy teacher (e.g., DeepSeek-R1) to guide updates. While works like GHPO [1] and RLPR [2] propose specific adaptive mechanisms or reward formulations, they are fundamentally designed for static distillation contexts or single-policy.
>
>     **Our Contribution:** We identify that relying on a single teacher, no matter how powerful, imposes a **"single-source limitation"** on the student's exploration, potentially restricting the diversity of reasoning paths and may introduce reasoning bias. AMPO adapts mechanisms from prior works (like adaptive triggering) specifically to solve this **Multi-Teacher Integration** challenge, creating a unified framework that dynamically selects the most "comprehensible" guidance from a diverse pool of peers.
>
> * **Empirical Justification:**
>     Our motivation is grounded in the **"Wisdom of Crowds"** and backed by our experimental results. As shown in the comparison below (extracted from the updated Table 1), utilizing **4 peer-sized teachers (AMPO)** consistently outperforms using a **single powerful teacher (DeepSeek-R1)** under identical budgets (8.5k samples) and settings. This advantage is evident not only in **Out-of-Distribution (OOD)** tasks but also in **In-Distribution (Math)** reasoning:
>
>     | Method | Teacher Composition | **In-Dist Avg. (Math)** | **Out-of-Dist Avg.** |
>     | :--- | :--- | :---: | :---: |
>     | **AMPO (R1 Only)** | Single Strong Teacher (DeepSeek-R1) | 38.0 | 58.6 |
>     | **AMPO (4 LongCoTs)** | 4 Peer-Sized Teachers | **40.4** (+2.4) | **68.5** (+9.9) |
>
>     This empirical evidence confirms that the **Multi-Teacher strategy** itself, rather than just the combination of sub-modules, is the primary driver of the significant performance gains ($+9.9\%$ on OOD).
>
> **2. Clarification on Teacher Log Probabilities (Weakness 2)**
>
> > *Requires the teacher log probabilities to calculate the importance sampling term, might be too restrictive for certain scenarios*
>
> We wish to clarify a potential misunderstanding regarding our implementation. **AMPO does NOT require access to teacher log probabilities during training.**
>
> * **Importance Sampling (IS):** In practice, for off-policy updates, we adopt the standard assumption in offline RL/distillation where the teacher's probability for the provided high-quality trace is treated as $\pi_{teacher}(o|q) = 1$. This allows us to use pre-collected demonstrations from any model (even black-box APIs) without needing their internal states.
> * **Comprehension-based Selection ($r_p$):** The "comprehension score" is calculated using the **student (policy) model's** log probabilities, not the teacher's. We compute the likelihood of the *student* generating the correct answer given the *teacher's* reasoning path.
> * **Deployment:** No teacher models are loaded during training. All guidance is retrieved from a static **Multi-Guidance Pool**.
> We will revise Sections 3.3 and 3.4 to explicitly state that teacher log probabilities are not required, ensuring the method is computationally efficient and widely applicable.

---

> ### Author Response · Authors · 2025-11-23
> **Response to Reviewer y1Nj (2/2)**
>
> **3. Comparisons on Smaller Models (Weakness 3)**
>
> > *The method needs several teacher models, but was only compared to GRPO as the baseline for llama-8b and qwen 1.5b, which is unfair.*
>
> We agree that a comprehensive baseline comparison is essential. In our initial submission, we prioritized the 7B experiments. Following your advice, we have conducted full **SFT** and **SFT+GRPO** experiments for **Llama-3.2-8B-Ins** and **Qwen2.5-1.5B-Ins** to verify generalizability.
>
> The updated results (detailed in **Table 1** below) demonstrate that AMPO achieves state-of-the-art performance across these models, outperforming both SFT and SFT+GRPO baselines on both In-Distribution and OOD benchmarks.
>
> **Updated Experimental Results (Table 1)**
>
> Below is the comprehensive performance comparison, including the newly added baselines for smaller models and the HumanEval benchmark.
>
> | Method | AIME 24/25 | AMC | MATH 500 | Minerva | Olympiad | **In-Dist Avg.** | ARC-c | GPQA* | MMLU Pro | Human Eval | **OOD Avg.** |
> | :--- | :---: | :---: | :---: | :---: | :---: | :---: | :---: | :---: | :---: | :---: | :---: |
> | **Qwen2.5-1.5B-Ins** | 2.8/1.3 | 21.9 | 51.4 | 19.1 | 19.1 | **19.3** | 42.3 | 0.5 | 25.1 | 42.1 | **27.5** |
> | SFT | 0.8/0.2 | 13.6 | 35.6 | 7.0 | 12.0 | **11.8** | 7.9 | 8.1 | 16.5 | 49.4 | **20.5** |
> | GRPO | 2.9/0.8 | 24.0 | 53.8 | 17.0 | 19.1 | **19.6** | 68.7 | 21.7 | 34.1 | 48.8 | **43.3** |
> | SFT+GRPO | 1.8/0.3 | 24.5 | 52.8 | 11.0 | 19.3 | **18.4** | 71.8 | 24.7 | 31.7 | 51.2 | **44.9** |
> | **AMPO (4LongCoTs)** | 4.9/1.9 | 28.5 | 58.2 | 17.6 | 23.9 | **22.5** | 73.5 | 27.8 | 34.1 | 50.0 | **46.3** |
> | | | | | | | | | | | | |
> | **Llama3.2-8B-Ins** | 2.8/0.4 | 20.2 | 45.4 | 19.1 | 14.5 | **17.1** | 44.3 | 0.0 | 40.2 | 59.8 | **36.1** |
> | SFT | 1.8/0.8 | 18.6 | 46.6 | 20.6 | 17.9 | **17.7** | 83.2 | 20.2 | 39.6 | 56.7 | **50.0** |
> | GRPO | 2.1/0.1 | 16.7 | 44.2 | 24.6 | 12.0 | **16.6** | 78.6 | 9.3 | 38.3 | 57.3 | **45.9** |
> | SFT+GRPO | 4.2/0.2 | 19.4 | 51.0 | 22.4 | 20.1 | **19.6** | 86.1 | 31.3 | 47.0 | 63.4 | **57.0** |
> | **AMPO (4LongCoTs)** | 9.4/1.7 | 26.9 | 59.6 | 37.9 | 25.7 | **24.5** | 88.0 | 29.8 | 52.8 | 68.9 | **59.9** |
> | | | | | | | | | | | | |
> | **Qwen-7B-Instruct** | 12.3/6.5 | 43.8 | 75.8 | 36.4 | 38.8 | **35.6** | 85.1 | 6.6 | 55.8 | 80.5 | **57.0** |
> | SFT | 12.7/14 | 41.0 | 76.0 | 30.5 | 36.4 | **35.1** | 80.0 | 17.2 | 45.2 | 82.9 | **56.3** |
> | GRPO | 11.3/9.9 | 46.6 | 76.8 | 34.6 | 37.6 | **36.1** | 92.1 | 5.1 | 58.7 | 74.4 | **57.6** |
> | SFT+GRPO | 20.1/16.7 | 53.4 | 81.0 | 35.0 | 50.0 | **42.7** | 91.0 | 38.4 | 58.7 | 83.5 | **67.9** |
> | LUFFY | 16.4/15.2 | 49.3 | 81.6 | 37.9 | 45.8 | **41.0** | 92.0 | 21.7 | 60.3 | 78.1 | **63.0** |
> | **AMPO (4LongCoTs)** | 14/14.1 | 48.0 | 80.8 | 39.3 | 45.9 | **40.4** | 92.7 | 40.4 | 59.6 | 81.1 | **68.5** |
> | AMPO (Only R1) | 14.8/9.2 | 48.8 | 77.0 | 35.7 | 42.5 | **38.0** | 92.9 | 0.5 | 59.7 | 81.1 | **58.6** |
> | SFT+AMPO | 19/18.3 | 53.2 | 82.0 | 37.5 | 49.5 | **43.2** | 90.7 | 38.4 | 56.7 | 82.3 | **67.0** |
>
> **4. Justification of the Shaping Function $f(x)$ (Question 1)**
>
> > *What is the justification of using the shaping function, f(x)?*
>
> The shaping function $f(x) = \frac{x}{x+0.1}$ is adopted from **LUFFY** ([Yan et al., 2025](https://arxiv.org/abs/2504.14945)). It serves as a **"filter and amplifier"** to prevent entropy collapse in Mixed-Policy RL. It encourages the model to focus on learning valid reasoning steps that the current policy might assign low probability to (i.e., "hard" steps verified by experts), rather than just imitating simple tokens it has already mastered.
> While we adopted this effective design from LUFFY, we improved upon it by implementing **sequence-level loss aggregation** (Section 3.4) instead of token-level. This modification is crucial for AMPO to ensure fair weighting across diverse teacher responses of varying lengths.

---

### Official Review · Reviewer_dXdg · 2025-10-25

**Soundness:** 2
**Presentation:** 3
**Contribution:** 2
**Rating:** 4
**Confidence:** 4

**Summary:**

This paper proposes AMPO (Adaptive Multi-Guidance Policy Optimization), a mixed-policy RL framework that leverages multiple teacher models to enhance reasoning capabilities in LLMs. The core contributions are: (1) an adaptive replacement mechanism that provides external guidance only when the on-policy model completely fails, and (2) a comprehension-based selection strategy that guides the student to learn from reasoning paths it is most likely to understand. Experiments on mathematical reasoning benchmarks show AMPO outperforms GRPO by 4.3% on in-distribution tasks and 12.2% on out-of-distribution tasks, while achieving comparable performance to single-teacher methods using significantly less data.

**Strengths:**

1. The paper clearly identifies limitations of single-teacher approaches in mixed-policy RL and provides convincing motivation for multi-teacher strategies.

2. The combination of adaptive triggering (guidance-on-demand) and comprehension-based selection is intuitive.

3. The 12.2% improvement on OOD tasks and efficient use of 8.5k vs 46k samples demonstrate practical value. The Pass@k and entropy analyses convincingly show enhanced exploration.

**Weaknesses:**

1. While empirically successful, the paper lacks theoretical analysis of why multiple peer-sized teachers should outperform a single stronger teacher. What are the theoretical conditions under which diversity trumps capability?

2. Experiments focus heavily on mathematical reasoning. Generalization to other domains (code, creative writing, general reasoning) remains unclear. The paper would benefit from at least one non-math domain.

3. I have some scalability concerns: Does the approach work with 10+ teachers? What about continually adding new teachers? How to select the optimal k0 automatically?

4. Missing comparisons: No comparison with other recent multi-teacher or ensemble RL methods. The baseline is primarily GRPO and one single-teacher method (LUFFY).

**Questions:**

1. Can you provide any theoretical analysis (e.g., regret bounds, convergence guarantees) showing when multi-teacher diversity is provably beneficial?

2. How does AMPO (4×7B teachers) compare against using DeepSeek-R1 or GPT-4 as a single teacher with the same 8.5k budget?

3. Have you considered learned selection mechanisms (e.g., using a meta-learner to choose teachers) instead of comprehension-based scoring?

4. When does AMPO underperform? Are there cases where single-teacher guidance is actually better?

5. Can you quantify teacher diversity (e.g., solution agreement rate, trajectory distance) and correlate it with performance gains?

---

> ### Author Response · Authors · 2025-11-23
> **Response to Reviewer dXdg (1/2)**
>
> We sincerely thank the reviewer for the detailed and constructive feedback. We appreciate your recognition of AMPO's "convincing motivation" and the "practical value" demonstrated by our 12.2% improvement on OOD tasks. Below, we address your concerns regarding theoretical analysis, generalization, and scalability.
>
> **1. Theoretical Analysis & "Diversity Trumps Capability" (Weakness 1, Question 1, 2)**
>
> > *While empirically successful, the paper lacks theoretical analysis of why multiple peer-sized teachers should outperform a single stronger teacher. What are the theoretical conditions under which diversity trumps capability?*
>
> >*Can you provide any theoretical analysis (e.g., regret bounds, convergence guarantees) showing when multi-teacher diversity is provably beneficial?*
>
> > *How does AMPO (4×7B teachers) compare against using DeepSeek-R1 or GPT-4 as a single teacher with the same 8.5k budget?*
>
> **Empirical Validation over Theoretical Bounds:**
> While we acknowledge that a formal theoretical derivation (e.g., regret bounds) is beyond the scope of this primarily empirical work, our results align with the established principle of the **"Wisdom of Crowds"**，as the proverb goes, *"Two heads are better than one"* (implying that the collective wisdom of ordinary people can exceed that of a single genius).
>
> Our experiments provide robust empirical evidence for this:
> 1.  **Data Efficiency:** AMPO (8.5k samples, 4 peer teachers) matches the performance of LUFFY (46k samples, 1 strong teacher R1).
> 2.  **Direct Comparison (Controlled Budget):** To directly answer your Question 2, we conducted a controlled experiment (detailed in Section 4.4 and the updated Table 1 below) where all settings (8.5k budget, training hyperparameters) were identical, varying only the teacher composition.
>
> | Method | Teacher Composition | **In-Dist Avg.** | **Out-of-Distribution Avg.** |
> | :--- | :--- | :---: | :---: |
> | **AMPO (R1 Only)** | Single Strong Teacher (DeepSeek-R1) | 38.0 | 58.6 |
> | **AMPO (4 LongCoTs)** | 4 Peer-Sized Teachers | **40.4** (+2.4) | **68.5** (+9.9) |
>
> **Conclusion:** The multi-teacher configuration significantly outperforms the single "super-teacher" baseline, especially on OOD tasks (+9.9%). This suggests that while a single strong teacher provides high-quality but potentially homogeneous reasoning paths, a diverse pool of peers provides a broader coverage of the solution space, preventing the student from overfitting to a single reasoning style.
>
> **2. Generalization to Non-Math Domains (Weakness 2)**
>
> > *Experiments focus heavily on mathematical reasoning. Generalization to other domains (code, creative writing, general reasoning) remains unclear. The paper would benefit from at least one non-math domain.*
>
> We respectfully clarify that our "Out-of-Distribution" (OOD) benchmarks already encompass broad non-math domains:
> * **ARC-c:** Tests common sense and logical reasoning (Grade 3-9 science).
> * **GPQA-diamond:** Covers expert-level biology, physics, and chemistry.
> * **MMLU-Pro:** A comprehensive suite covering 14 categories including Humanities, Social Sciences, and Engineering.
>
> Furthermore, following your suggestion, we evaluated all our models on the **HumanEval (Code Generation)** benchmark. The results, presented below, demonstrate that AMPO achieves competitive or superior performance across different model families, proving its effectiveness extends to coding tasks.
>
> **Table: HumanEval (Pass@1) Performance across Model Families**
> | Method | **Qwen2.5-1.5B-Ins** | **Llama3.2-8B-Ins** | **Qwen2.5-7B-Instruct** |
> | :--- | :---: | :---: | :---: |
> | Base Model | 42.1 | 59.8 | 80.5 |
> | SFT | 49.4 | 56.7 | 82.9 |
> | GRPO | 48.8 | 57.3 | 74.4 |
> | SFT+GRPO | **51.2** | 63.4 | **83.5** |
> | **AMPO (Ours)** | 50.0 | **68.9** | 81.1 |
>
> Notably, on **Llama3.2-8B-Ins**, AMPO achieves a remarkable **+11.6%** improvement over the GRPO baseline, further validating its robust generalization capability.

---

> ### Author Response · Authors · 2025-11-23
> **Response to Reviewer dXdg (2/2)**
>
> **3. Scalability and Teacher Selection (Weakness 3, Question 3)**
>
> > *I have some scalability concerns: Does the approach work with 10+ teachers? What about continually adding new teachers?*
>
> > *Have you considered learned selection mechanisms (e.g., using a meta-learner to choose teachers) instead of comprehension-based scoring?*
>
> **Scalability:**
> Yes, AMPO is theoretically scalable to 10+ teachers. Our largest experiment used 8 teachers. The primary cost of adding teachers is in the **Guidance Selection** phase, where we compute the likelihood (logits) of the ground truth given the teacher's reasoning. This inference cost is linear with respect to the number of teachers but is computationally much cheaper than the training itself. Therefore, scaling to a larger pool is feasible.
>
> **Meta-Learner vs. Comprehension Metric:**
> We agree that a learned meta-selector is an intriguing direction. However, we chose the **Comprehension-based metric**($r_p$) for two reasons:
> 1.  **Adaptability:** It requires no training. A meta-learner would need to be trained for each specific student-teacher pair, whereas $r_p$ is a zero-shot metric that automatically adapts as the student model evolves.
> 2.  **Efficiency:** It avoids the computational overhead of training and maintaining a separate selection network.
>
> We appreciate the helpful suggestion and will pursue targeted experiments to explore this direction in the future.
>
> **4. Dynamic $k_0$ Heuristic (Weakness 3)**
>
> > *How to select the optimal $k_0$ automatically?*
>
> Currently, we use a fixed $k_0$. However, we are exploring a **Dynamic Balancing Strategy**: we aim to maintain a balanced ratio of correct-to-incorrect signals in each batch. Specifically, if the student's self-generated correct rollout count ($N_{correct}$) is less than 50% of the rollout size ($G \times 0.5$), we dynamically set $k = (G \times 0.5) - N_{correct}$. This ensures a consistent learning signal without manual tuning.
>
> We have already initiated these experiments, if they validate our conjecture, we will either present the findings in the discussion section or revise the paper’s principal methodology to reflect the updated approach.
>
> **5. Underperformance and Response Length (Question 4)**
>
> > *When does AMPO underperform? Are there cases where single-teacher guidance is actually better?*
>
> Based on our experiments, AMPO consistently outperforms baselines in accuracy. The primary trade-off is **Response Length** when compared to pure on-policy RL.
>
> As shown in **Table 2** of our paper, AMPO induces a longer average response length (**1268 tokens**) compared to **GRPO (881 tokens)**. This increase reflects the cost of "deeper thinking" and assimilating external teacher guidance. However, it is crucial to note that AMPO is significantly more efficient than **SFT (5962 tokens)** or **SFT+GRPO (3318 tokens)**. Thus, while AMPO incurs a length cost compared to the concise (but often less accurate) GRPO, it strikes a much better balance between efficiency and performance than standard SFT approaches.
>
> **6. Diversity Quantification (Question 5)**
>
> > *Can you quantify teacher diversity (e.g., solution agreement rate, trajectory distance) and correlate it with performance gains?*
>
> In this work, we relied on the extrinsic evaluation (performance gain) as the proxy for diversity's benefit. We agree that intrinsic metrics, such as the semantic distance between reasoning trajectories, would provide deeper insights. We view this quantitative interpretability analysis as a promising avenue for future research.

---

> ### Author Response · Authors · 2025-11-26
> **Follow-up: Experimental Verification of Automatic k_0 Selection(Weakness 3)**
>
> Dear Reviewer dXdg,
>
> In our previous response regarding your question on **"how to select the optimal $k_0$ automatically,"** we proposed a hypothesis centered on dynamically balancing the ratio of correct samples. We have now verified this with supplementary experiments.
>
> **1. Methodology: Automatic Balancing**
>
> We replaced the manual hyperparameter $k_0$ with an automated logic.
> * **Original (Sparse Trigger):** Replacement occurred only if $N_{correct} = 0$.
> * **New (Balanced Trigger):** Replacement occurs whenever the model's self-exploration yields insufficient correct solutions, specifically when $N_{correct} < G/2$ (where $G$ is the rollout size).
>
> The system then automatically determines the replacement count $k$ to reach the 50% target, subject to the limitation of the Multi-Guidance Pool ($N_g$):
> $$k = \min\left(\frac{G}{2} - N_{correct}, N_g\right)$$
>
> This allows the framework to define the "optimal" $k$ on a per-sample basis automatically.
>
> **2. Comparative Results**
>
> | Setting                  |   AIME 24/25    |   AMC    | MATH 500 | Minerva  | Olympiad | **Avg. Score** | **Avg. Length** |
> | :----------------------- | :-------------: | :------: | :------: | :------: | :------: | :------------: | :-------------: |
> | Fixed $k_0=1$            |   16.3 / 11.0   |   50.2   |   78.2   |   36.8   |   40.6   |      38.8      |      1104       |
> | Fixed $k_0=2$            |   17.5 / 12.2   |   52.6   |   80.6   |   37.1   |   46.1   |      41.0      |      2439       |
> | Fixed $k_0=3$            |   16.9 / 12.5   |   51.2   |   81.2   |   37.9   |   47.0   |      41.1      |      2103       |
> | Fixed $k_0=4$            |   16.6 / 15.3   |   51.3   |   82.2   |   38.2   |   45.3   |      41.5      |      1750       |
> | **Automatic (Adaptive)** | **17.4 / 17.7** | **50.5** | **80.2** | **39.0** | **44.7** |    **41.6**    |    **1234**     |
>
> The experiment confirms that this automated strategy is superior to manual tuning:
> * It yields the **highest overall performance (41.6%)**.
> * It drastically optimizes efficiency (**Avg. Length: 1234**), avoiding the unnecessary computational cost observed in fixed $k_0 \ge 2$ settings.
>
> We are grateful for your insightful question, which drove us to validate this automated selection mechanism. We will detail this formulation and result in the **Discussion** of the final revision.

---

### Official Review · Reviewer_Lxdg · 2025-10-31

**Soundness:** 3
**Presentation:** 3
**Contribution:** 2
**Rating:** 4
**Confidence:** 3

**Summary:**

The paper proposes Adaptive Multi-Guidance Policy Optimization (AMPO), a new reinforcement-learning framework designed to enhance the reasoning capability of large language models (LLMs) through multi-teacher guidance. Existing Reinforcement Learning with Verifiable Rewards (RLVR) methods—such as Group Relative Policy Optimization (GRPO)—often rely on a single off-policy teacher or pure self-exploration, limiting the diversity of reasoning paths explored. AMPO overcomes this limitation by (1) selectively invoking multiple teachers only when the model fails (the adaptive replacement mechanism), and (2) ranking external guidance using a comprehension-based selection metric derived from the student model’s likelihood of reproducing the teacher’s reasoning. The approach demonstrates consistent improvements—4.3% on in-distribution and 12.2% on out-of-distribution benchmarks—while maintaining training stability and higher reasoning diversity

**Strengths:**

The paper introduces a well-motivated and clearly structured framework that generalizes the idea of mixed-policy RL with an elegant “guidance-on-demand” paradigm.

AMPO improves both in-distribution (math reasoning) and out-of-distribution generalization, outperforming GRPO and matching or surpassing LUFFY with 5× less data.

Robust performance across multiple model sizes (1.5B, 7B, 8B) and architectures (Qwen, LLaMA) underscores scalability.

**Weaknesses:**

While AMPO is compared against LUFFY and GRPO, there is no direct comparison with multi-teacher SFT frameworks (e.g., TWT 2025 (https://arxiv.org/abs/2503.24198) or multi-distillation baselines). This makes it difficult to isolate the incremental benefits of AMPO’s adaptive mechanism over simple multi-teacher data aggregation.

The experiments show that mixing LongCoT and ShortCoT teachers affects both accuracy and reasoning length. However, the criteria for teacher selection and diversity quantification remain heuristic. A deeper ablation or diversity metric (e.g., representation distance, reasoning style variance) would strengthen the argument.

**Questions:**

None

---

> ### Author Response · Authors · 2025-11-23
> **Response to Reviewer Lxdg**
>
> We sincerely thank the reviewer for the thoughtful feedback and for recognizing AMPO as a "well-motivated," "clearly structured," and "elegant" framework. We are pleased that you acknowledge the robustness of our approach across multiple model sizes. Below, we address your concerns regarding the baselines and diversity metrics.
>
> **1. Comparison with Multi-Teacher SFT Frameworks**
>
> > *While AMPO is compared against LUFFY and GRPO, there is no direct comparison with multi-teacher SFT frameworks (e.g., TWT 2025 (https://arxiv.org/abs/2503.24198) or multi-distillation baselines). This makes it difficult to isolate the incremental benefits of AMPO’s adaptive mechanism over simple multi-teacher data aggregation.*
>
> We appreciate this opportunity to clarify our baseline setup. **We would like to highlight that the "SFT" baseline reported in our paper is, in fact, a Multi-Teacher SFT (Multi-Distillation) baseline.**
>
> As detailed in **Appendix A.1.2**, our SFT dataset was constructed by aggregating all correct reasoning paths generated by the four diverse teachers (AceReason, DeepSeek-R1-Distill, OpenR1, and Qwen3). This process yielded a comprehensive dataset of **32,252 samples**, where each problem is paired with 1 to 4 diverse correct demonstrations.
>
> Therefore, the comparison between **SFT** and **AMPO** in our experiments (Table 1) **is specifically designed** to isolate the incremental benefits of our adaptive RL mechanism over simple multi-teacher data aggregation.
>
> The results (updated with our latest experiments on smaller models) clearly demonstrate this benefit:
>
> | Method | **In-Distribution Avg.** | **Out-of-Distribution Avg.** |
> | :--- | :---: | :---: |
> | **Qwen2.5-1.5B-Ins** | | |
> | SFT (Multi-Teacher) | 11.8 | 20.5 |
> | **AMPO** | **22.5** (+10.7) | **46.3** (+25.8) |
> | **Llama3.2-8B-Ins** | | |
> | SFT (Multi-Teacher) | 17.7 | 50.0 |
> | **AMPO** | **24.5** (+6.8) | **59.9** (+9.9) |
> | **Qwen2.5-7B-Instruct** | | |
> | SFT (Multi-Teacher) | 35.1 | 56.3 |
> | **AMPO** | **40.4** (+5.3) | **68.5** (+12.2) |
>
> *(Note: The SFT results above utilize the full multi-teacher dataset and the OOD benchmark has added test results for HumanEval. AMPO consistently outperforms this strong multi-distillation baseline, proving that the adaptive guidance mechanism in RL provides significant value beyond standard supervised learning.)*
>
> We will explicitly label the SFT baseline as "Multi-Teacher SFT" in the revised manuscript to prevent future confusion.
>
> **Regarding TWT 2025 and Related Work**
>
> We acknowledge the relevance of TWT (https://arxiv.org/abs/2503.24198). TWT primarily focuses on constructing efficient datasets for **Supervised Fine-Tuning (SFT)** to enhance small model concise reasoning. Since the code for TWT is not open-source, we were unable to reproduce it exactly. However, our "Multi-Teacher SFT" baseline conceptually aligns with the core idea of TWT, using filtered multi-teacher data to enhance supervision.
>
> The key distinction of our work is the integration of this concept into **Reinforcement Learning**. While most concurrent works in the RLVR paradigm rely on a single teacher or self-exploration, AMPO is one of the first frameworks to explore an end-to-end **Multi-Teacher RL** approach. Our results show that **dynamic interaction with multiple teachers** during RL offers superior performance compared to static SFT aggregation alone.
>
> **2. Teacher Selection and Diversity Metrics**
>
> > *The experiments show that mixing LongCoT and ShortCoT teachers affects both accuracy and reasoning length. However, the criteria for teacher selection and diversity quantification remain heuristic. A deeper ablation or diversity metric (e.g., representation distance, reasoning style variance) would strengthen the argument.*
>
> We agree with the reviewer that a theoretical quantification of diversity (e.g., representation distance or reasoning style variance) is an exciting direction.
> In this work, our primary focus was to establish the empirical effectiveness of the AMPO framework. We adopted the comprehension-based selection ($r_p$) and response length heuristic as a practical and computationally efficient proxy for "useful diversity", prioritizing reasoning paths that the student model can actually absorb (high likelihood) and that are efficient (short length).
>
> Our ablation studies (Table 3 in the paper) confirm that **this heuristic significantly outperforms random selection**. While we did not include a theoretical diversity metric in this iteration, we believe our empirical results provide strong evidence for the method's validity. We view **the development of rigorous mathematical metrics for "reasoning diversity" in RL as a valuable topic for future research in the interpretability and explanation field**, and we hope our work inspires further exploration in this direction.

---

### Official Review · Reviewer_kg4R · 2025-11-01

**Soundness:** 3
**Presentation:** 2
**Contribution:** 2
**Rating:** 4
**Confidence:** 3

**Summary:**

This paper proposes a new algorithm, Adaptive Multi-Guidance Policy Optimization (AMPO), which enhances reasoning in LLMs by leveraging multiple teacher models when the generated results are incorrect. A comprehension-based guidance selection further filters off-policy traces so that the student agent learns from reasoning paths it can better understand. The authors validate the method’s efficiency through extensive experiments and ablation studies. Results show that AMPO achieves consistent gains, particularly on out-of-distribution benchmarks, while maintaining higher entropy during training, indicating stronger exploratory behavior.

**Strengths:**

The proposed method consistently improves over GRPO, especially on OOD tasks, demonstrating stronger generalization capability. AMPO achieves competitive or superior results using multiple weaker teachers, compared to single stronger teachers or self-exploration, showing improved data efficiency. The paper includes detailed ablation studies on various components of AMPO, providing clear intuition on how adaptive replacement and comprehension-based selection jointly maintain exploration and learning efficiency.

**Weaknesses:**

1. Constructing the 8.5k multi-teacher verified dataset appears computationally intensive. More detail on the associated cost or resources would help contextualize the efficiency and scalability claims.
2. Baseline Inconsistency:
 For Qwen 2.5-1.5B and LLaMA 3.2-8B, only GRPO is used as a baseline, while the 7B model compares against SFT, SFT + GRPO, and LUFFY.  Including at least SFT or SFT + GRPO for the smaller models would give a fairer picture. Additionally, the 0 % GPQA-diamond result for LLaMA 3.2 8B seems unexpectedly low and should be re-checked.
3. The same symbol \hat{A} is used for both on-policy and off-policy adavantages, though they are computed over different sample sets. It would help to introduce explicit subscripts (e.g., \hat{A}^{on}, \hat{A}^{off}) to avoid ambiguity.
4. Minor issue:
   1. The numerical result reported around Line 341 does not match Table 2.

**Questions:**

1. The ablation shows that larger $k_0$ leads to higher accuracy but also longer responses. Could the authors provide practical guidance or heuristics for selecting $k_0$, for example, balancing training variance against computational budget?

---

> ### Author Response · Authors · 2025-11-23
> **Response to Reviewer kg4R (1/2)**
>
> We genuinely thank the reviewer for the constructive feedback and the positive assessment of our work. We are encouraged that you find AMPO to be a sound approach that demonstrates strong generalization capabilities. Below, we address your concerns point-by-point.
>
> **1. Computational Cost of Dataset Construction**
>
> > *Of course, this is only our hypothesis, we are currently conducting experiments to evaluate this dynamic heuristic strategies, and if the results are positive we will report them in the paper’s discussion section or incorporate them into a revised description of the main methodology.*
>
> Thank you for this suggestion. We agree that transparency regarding resource usage is vital.
> To construct the multi-teacher dataset, we utilized **4 NVIDIA RTX 5090 GPUs** over a period of approximately **one to two weeks**. This process involved generating responses from multiple teachers and filtering them via the verifier. We will add a detailed breakdown of these computational resources and the time cost in the **Appendix** of the final revision to provide a clearer context for future reproducibility and scalability comparisons.
>
> **2. Baseline Inconsistency & GPQA-diamond Results**
>
> > *Baseline Inconsistency: For Qwen 2.5-1.5B and LLaMA 3.2-8B, only GRPO is used as a baseline, while the 7B model compares against SFT, SFT + GRPO, and LUFFY. Including at least SFT or SFT + GRPO for the smaller models would give a fairer picture. Additionally, the 0 % GPQA-diamond result for LLaMA 3.2 8B seems unexpectedly low and should be re-checked.*
>
> We sincerely apologize for the initial omission of these baselines due to time constraints. Following your advice, we have conducted comprehensive experiments adding **SFT** and **SFT+GRPO** baselines for both Qwen2.5-1.5B-Ins and LLaMA3.2-8B-Ins. The updated results are presented in the table below. This data will be used to update the experimental results (Table 1) in our revised paper.
>
> | Method | AIME 24/25 | AMC | MATH 500 | Minerva | Olympiad | **In-Dist Avg.** | ARC-c | GPQA* | MMLU Pro | Human Eval | **OOD Avg.** |
> | :--- | :---: | :---: | :---: | :---: | :---: | :---: | :---: | :---: | :---: | :---: | :---: |
> | **Qwen2.5-1.5B-Ins** | 2.8/1.3 | 21.9 | 51.4 | 19.1 | 19.1 | **19.3** | 42.3 | 0.5 | 25.1 | 42.07 | **27.5** |
> | SFT | 0.8/0.2 | 13.6 | 35.6 | 7.0 | 12.0 | **11.8** | 7.9 | 8.1 | 16.5 | 49.39 | **20.5** |
> | GRPO | 2.9/0.8 | 24.0 | 53.8 | 17.0 | 19.1 | **19.6** | 68.7 | 21.7 | 34.1 | 48.78 | **43.3** |
> | SFT+GRPO | 1.8/0.3 | 24.5 | 52.8 | 11.0 | 19.3 | **18.4** | 71.8 | 24.7 | 31.7 | 51.22 | **44.9** |
> | **AMPO (ours)** | 4.9/1.9 | 28.5 | 58.2 | 17.6 | 23.9 | **22.5** | 73.5 | 27.8 | 34.1 | 50.0 | **46.3** |
> | | | | | | | | | | | | |
> | **Llama3.2-8B-Ins** | 2.8/0.4 | 20.2 | 45.4 | 19.1 | 14.5 | **17.1** | 44.3 | 0.0 | 40.2 | 59.76 | **36.1** |
> | SFT | 1.8/0.8 | 18.6 | 46.6 | 20.6 | 17.9 | **17.7** | 83.2 | 20.2 | 39.6 | 56.71 | **50.0** |
> | GRPO | 2.1/0.1 | 16.7 | 44.2 | 24.6 | 12.0 | **16.6** | 78.6 | 9.3 | 38.3 | 57.32 | **45.9** |
> | SFT+GRPO | 4.2/0.2 | 19.4 | 51.0 | 22.4 | 20.1 | **19.6** | 86.1 | 31.3 | 47.0 | 63.41 | **57.0** |
> | **AMPO (ours)** | 9.4/1.7 | 26.9 | 59.6 | 37.9 | 25.7 | **24.5** | 88.0 | 29.8 | 52.8 | 68.9 | **59.9** |
>
> **Regarding the 0% GPQA score:**
> We re-evaluated the LLaMA3.2-8B-Ins model (both Base and GRPO versions) on the GPQA-diamond dataset three times to ensure accuracy. The detailed results are as follows:
>
> | Method | GPQA*_test1 | GPQA*_test2 | GPQA*_test3 | **Avg** |
> | :--- | :---: | :---: | :---: | :---: |
> | Llama3.2-8B-Ins | 0 | 0 | 0 | **0** |
> | GRPO | 9.6 | 9.6 | 8.6 | **9.3** |
>
> As shown, the **Base model** indeed scores **0%**, likely due to the extreme difficulty of the questions for this model. However, after re-testing, the **GRPO** baseline shows an average of **9.3%** (corrected from 0% in our initial submission).
>
> Crucially, **AMPO achieves 29.8%**, and **SFT+GRPO reaches 31.3%**. While SFT+GRPO is not our primary method, its high performance validates the quality of our **Multi-Guidance Pool**, as the SFT dataset was constructed using the exact same multi-teacher data pipeline used in AMPO (detailed in **Appendix A.1.2**). This confirms that **incorporating diverse teacher perspectives whether via AMPO or SFT initialization could significantly unlock the model's potential and generalization capabilities** compared to standard GRPO (9.3%).

---

> ### Author Response · Authors · 2025-11-23
> **Response to Reviewer kg4R (2/2)**
>
> **3. Mathematical Notation Ambiguity**
>
> > *The same symbol  $\hat{A}$  is used for both on-policy and off-policy advantages, though they are computed over different sample sets. It would help to introduce explicit subscripts (e.g., $\hat{A}^{on}$, $\hat{A}^{off}$) to avoid ambiguity.*
>
> Thank you for pointing this out. In the original text, we differentiated them implicitly via the response indices $i$ (on-policy) and $j$ (off-policy). We agree that explicit notation is clearer. In the revision, we will adopt your suggestion and denote them as $\hat{A}^{on}$ and $\hat{A}^{off}$ with the corresponding indices($i$ or $j$) to rigorously distinguish between advantages computed from on-policy samples versus off-policy guidance.
>
> **4. Numerical Mismatch**
>
> > *The numerical result reported around Line 341 does not match Table 2.*
>
> We appreciate your attention to detail. This was a typographical error. We will correct the text to ensure it strictly matches the data reported in Table 2 in the final version.
>
> **Q1. Practical Guidance for Selecting $k_0$**
>
> > *The ablation shows that larger $k_0$ leads to higher accuracy but also longer responses. Could the authors provide practical guidance or heuristics for selecting $k_0$, for example, balancing training variance against computational budget?*
>
> Thanks for the insightful question. In our main experiments, we used a fixed $k_0$. However, based on our analysis of the trade-off between accuracy and response length/training stability, we hypothesize that a **dynamic $k_0$ strategy** would be the optimal heuristic.
>
> Specifically, for a rollout size of $G$ (e.g., $G=8$), we suggest maintaining a balanced ratio of correct-to-incorrect signals. A potential heuristic is to dynamically set $k$ such that the total number of correct responses (on-policy successes + external guidance) equals roughly $50\%$ of the batch size ($G/2$).
> * If the model generates $N_{correct}$ correct responses and $N_{correct} < G/2$, we set $k = G/2 - N_{correct}$.
> * This ensures the model always has sufficient positive reinforcement without overwhelming the training with purely off-policy data.
>
> Of course, this is only our hypothesis, we are currently conducting experiments to evaluate this dynamic heuristic strategies, and if the results are positive we will report them in the paper’s discussion section or incorporate them into a revised description of the main methodology.

---

> ### Author Response · Authors · 2025-11-26
> **Follow-up Experiment: Validation of Dynamic k_0 Heuristic(Q1)**
>
> Dear Reviewer kg4R,
>
> Following up on our previous discussion regarding **"practical guidance or heuristics for selecting $k_0$"**, we have completed the experiments for the **Dynamic $k_0$ Strategy** (Adaptive Balancing). We are excited to share these results, which confirm that a dynamic approach outperforms fixed settings.
>
> **1. Refined Mechanism: From "Sparse" to "Balanced" Trigger**
>
> In our original paper (Eq. 3), the replacement flag $I$ was triggered only when *all* responses failed ($\forall i, R(o_i) < \tau$).
> To implement the **Dynamic Heuristic**, we modified this logic to target a balanced ratio (50% correct). The new replacement condition becomes:
>
> $I$ = $True$  if  $N_{correct} < \frac{G}{2}$ else  $False$
>
> where $N_{correct}$ is the number of correct on-policy responses in a rollout of size $G$.
> When triggered ($I=\text{True}$), the number of replacements $k$ is calculated to fill the gap, constrained by the availability of valid guidance ($N_g$) in the pool:
>
> $$k = \min\left(\frac{G}{2} - N_{correct}, N_g\right)$$
>
> This mechanism **aims to maintain** a balanced positive/negative sample ratio, rather than strictly ensuring it (as $N_g$ varies per problem).
>
> **2. Experimental Results**
>
> | $k_0$ Strategy         |   AIME 24/25    |   AMC    | MATH 500 | Minerva  | Olympiad | **Avg. Score** | **Avg. Length** |
> | :--------------------- | :-------------: | :------: | :------: | :------: | :------: | :------------: | :-------------: |
> | Fixed $k_0=1$          |   16.3 / 11.0   |   50.2   |   78.2   |   36.8   |   40.6   |      38.8      |      1104       |
> | Fixed $k_0=2$          |   17.5 / 12.2   |   52.6   |   80.6   |   37.1   |   46.1   |      41.0      |      2439       |
> | Fixed $k_0=3$          |   16.9 / 12.5   |   51.2   |   81.2   |   37.9   |   47.0   |      41.1      |      2103       |
> | Fixed $k_0=4$          |   16.6 / 15.3   |   51.3   |   82.2   |   38.2   |   45.3   |      41.5      |      1750       |
> | **Adaptive (Dynamic)** | **17.4 / 17.7** | **50.5** | **80.2** | **39.0** | **44.7** |    **41.6**    |    **1234**     |
>
> The results strongly validate this heuristic:
> * **Best Performance:** It achieves the highest average accuracy (**41.6%**).
> * **Efficiency:** It maintains a concise average response length (**1234 tokens**), significantly lower than high fixed settings (e.g., $k_0=2$ uses 2439 tokens).
>
> We sincerely thank you for raising this question. It inspired us to formalize this dynamic logic, which we will incorporate into the **Discussion** section of our revision and adopt as a core component in the future.

---

### Author Response · Authors · 2025-12-02
**Global Response to PCs, SACs, ACs and all of our Reviewers (1/3)**

Dear PCs, SACs, ACs and all of our Reviewers,

We sincerely appreciate the time and effort you have dedicated to reviewing our paper. Your constructive feedback has been invaluable in refining our work.

We are encouraged that the reviewers recognized the value of our work:
* **On Motivation & Novelty:** Reviewer Lxdg praised the framework as **"well-motivated and clearly structured"** with an **"elegant 'guidance-on-demand' paradigm."** Reviewer dXdg noted the paper provides **"clearly identifies limitations of single-teacher approaches."** and **"convincing motivation for multi-teacher strategies."**
* **On Effectiveness & Generalization:** Reviewer kg4R highlighted the **"stronger generalization capability"** and **"consistent gains, particularly on out-of-distribution benchmarks."** Reviewer dXdg acknowledged the **"practical value"** demonstrated by significant OOD improvements.
* **On Efficiency & Scalability:** Reviewer Lxdg highlighted that AMPO achieves superior results while **"matching or surpassing LUFFY with 5x less data,"** and noted that the **"performance across multiple model sizes (1.5B, 7B, 8B) and architectures (Qwen, LLaMA) underscores scalability."** Reviewer kg4R also commended the method for showing **"improved data efficiency."**
* **On Analysis & Design:** Reviewer dXdg found the combination of adaptive triggering and comprehension-based selection to be **"intuitive,"** and noted that the **"Pass@k and entropy analyses convincingly show enhanced exploration."** Reviewer kg4R appreciated the detailed ablation studies, which provide **"clear intuition"** on the method's components.

During the rebuttal period, we have made every effort to address the reviewers' concerns by **conducting extensive new experiments**, **refining our methodology**, and **revising the manuscript**. Below is a summary of our revisions.

### 1. Comprehensive Baseline Comparisons & Generalization
*(Concerns from Reviewers kg4R, y1Nj, dXdg)*

To ensure a rigorous evaluation, we have significantly expanded our experimental scope. The updated results are summarized in **Table 1** .

* **Full Baselines for Smaller Models:** We conducted full **SFT** and **SFT+GRPO** experiments for **Qwen2.5-1.5B-Ins** and **Llama-3.2-8B-Ins**. The results confirm that AMPO consistently outperforms robust baselines across all model scales.
* **Broad Generalization & Code:** We clarified that our existing OOD benchmarks (**ARC-c**, **GPQA***, **MMLU-Pro**) already encompass general reasoning, scientific inference, and multidisciplinary domains. To further verify that our math-centric training does not compromise general capabilities, we added **HumanEval** results. As shown in **Table 1**, AMPO maintains robust coding performance (**81.1%** for 7B), proving strong cross-domain generalization.
* **Unlocking Capabilities on GPQA\***: We addressed the query regarding the 0% score for the Llama-3.2-8B-Ins model on GPQA*. We verified this baseline performance and demonstrated that **AMPO boosts this to 29.8%**, highlighting a remarkable capability to unlock reasoning in challenging domains where base models fail.

### 2. Methodological Advancement: Dynamic $k_0$ Strategy
*(Heuristic concerns from Reviewers kg4R, dXdg)*

In response to questions about selecting the optimal guidance replacement number ($k_0$), we moved beyond the fixed hyperparameter approach. We hypothesized that a **Dynamic $k_0$ Balancing Strategy** would be optimal.

* **Mechanism:** This strategy dynamically calculates $k$ to maintain a balanced ratio of correct-to-incorrect responses (target: 50% correct) within a batch. We modified the trigger condition ($I$) and the replacement count ($k$) as follows:
    $$I = \begin{cases} \text{True} & \text{if } N_{correct} < \frac{G}{2} \\ \text{False} & \text{otherwise} \end{cases}$$
    $$k = \min(N_{g}, G/2 - N_{correct})$$
    where $N_{correct}$ is the number of correct on-policy responses, $G$ is the rollout size, and $N_{g}$ is the available guidance.

* **Superior Results:** As shown in the **Table 2**, this dynamic strategy achieves the **highest average accuracy (41.6%)** while maintaining a highly efficient response length (**1234 tokens**), effectively solving the trade-off between performance and training stability observed with fixed $k_0$.

---

> ### Author Response · Authors · 2025-12-02
> **Global Response to PCs, SACs, ACs and all of our Reviewers (2/3)**
>
> ### 3. Novelty & Empirical Validation: Wisdom of Crowds in Mixed-Policy RL
> *(Novelty and motivation concerns from Reviewers y1Nj, dXdg, Lxdg)*
>
> We clarified that AMPO is not merely a combination of existing modules but a novel framework designed to address the **"single-source limitation"** inherent in prior Mixed-Policy RL works (e.g., LUFFY), which rely on a single teacher. AMPO pioneers the integration of **"Wisdom of Crowds"** into Mixed-Policy RL to enhance exploratory diversity.
>
> * **Diverse Peers vs. Single Expert:** To validate this paradigm, we added a direct comparison between **AMPO (4 Peer Teachers)** and **AMPO (R1 Only)** under identical budgets. As shown below, the multi-teacher setup outperforms the single strong teacher, empirically proving that **diversity from peer models yields better gradients than a single expert source**.
>
>     | Method                | Teacher Composition       | In-Dist Avg. | OOD Avg. |
>     | :-------------------- | :------------------------ | :----------: | :------: |
>     | **AMPO (R1 Only)**    | Single Strong Teacher     |     38.0     |   58.6   |
>     | **AMPO (4 LongCoTs)** | **4 Peer-Sized Teachers** |   **40.4**   | **68.5** |
>
> * **SFT Clarification:** We clarified that our "SFT" baseline is effectively a **Multi-Teacher SFT (Multi-Distillation)** baseline constructed from our diverse pool. AMPO’s significant lead (+5.3% on In-Dist) isolates the specific benefits of our adaptive RL framework over static distillation.
>
> ### 4. Clarifications and Resource Transparency
> *(Technical details from Reviewers kg4R, y1Nj)*
>
> * **Computational Cost:** We added a detailed breakdown in **Appendix A.1.2**, specifying that dataset construction utilized **4 NVIDIA RTX 5090 GPUs** over approximately one week.
> * **Notation & Assumptions:** We resolved notation ambiguities ($\hat{A}^{on}$ vs. $\hat{A}^{off}$) and clarified that AMPO does not require teacher log probabilities during training (using the standard assumption $\pi_{teacher} \approx 1$), ensuring the method is computationally efficient and compatible with black-box teachers.
> * **Shaping Function $f(x)$:** We clarified that the shaping function serves as a **"filter and amplifier"** to to encourage the model to learn from high-quality off-policy traces that the current policy might assign low probability (adopted from LUFFY). We uniquely improved upon this by implementing **sequence-level aggregation** (Section 3.4) to ensure fair weighting across diverse teacher responses of varying lengths.
> * **Future Directions:** We expanded the **Limitations** section to acknowledge the need for theoretical diversity metrics and to discuss the scalability of our framework to larger teacher pools.
>
> Finally, we thank the reviewers for their constructive comments. We believe these additional results and clarifications firmly establish **AMPO** as a **data-efficient** and **scalable** framework for **complex LLM reasoning**. By intelligently leveraging **adaptive multi-teacher guidance**, AMPO effectively overcomes the limitations of single-teacher approaches, paving the way for more diverse exploration and robust generalization.
>
> Thank you again for your time and efforts.
>
> Best regards,
>
> The Authors of Submission 3689

---

> ### Author Response · Authors · 2025-12-02
> **Global Response to PCs, SACs, ACs and all of our Reviewers (3/3)**
>
> **Table 1: Updated Overall Performance (In-Distribution & Out-of-Distribution)**
> | Method               | AIME 24/25 |  AMC  | MATH 500 | Minerva | Olympiad | **In-Dist Avg.** | ARC-c | GPQA* | MMLU Pro | Human Eval | **OOD Avg.** |
> | :------------------- | :--------: | :---: | :------: | :-----: | :------: | :--------------: | :---: | :---: | :------: | :--------: | :----------: |
> | **Qwen2.5-1.5B-Ins** |  2.8/1.3   | 21.9  |   51.4   |  19.1   |   19.1   |     **19.3**     | 42.3  |  0.5  |   25.1   |    42.1    |   **27.5**   |
> | SFT                  |  0.8/0.2   | 13.6  |   35.6   |   7.0   |   12.0   |     **11.8**     |  7.9  |  8.1  |   16.5   |    49.4    |   **20.5**   |
> | GRPO                 |  2.9/0.8   | 24.0  |   53.8   |  17.0   |   19.1   |     **19.6**     | 68.7  | 21.7  |   34.1   |    48.8    |   **43.3**   |
> | SFT+GRPO             |  1.8/0.3   | 24.5  |   52.8   |  11.0   |   19.3   |     **18.4**     | 71.8  | 24.7  |   31.7   |    51.2    |   **44.9**   |
> | **AMPO (4LongCoTs)** |  4.9/1.9   | 28.5  |   58.2   |  17.6   |   23.9   |     **22.5**     | 73.5  | 27.8  |   34.1   |    50.0    |   **46.3**   |
> |                      |            |       |          |         |          |                  |       |       |          |            |              |
> | **Llama3.2-8B-Ins**  |  2.8/0.4   | 20.2  |   45.4   |  19.1   |   14.5   |     **17.1**     | 44.3  |  0.0  |   40.2   |    59.8    |   **36.1**   |
> | SFT                  |  1.8/0.8   | 18.6  |   46.6   |  20.6   |   17.9   |     **17.7**     | 83.2  | 20.2  |   39.6   |    56.7    |   **50.0**   |
> | GRPO                 |  2.1/0.1   | 16.7  |   44.2   |  24.6   |   12.0   |     **16.6**     | 78.6  |  9.3  |   38.3   |    57.3    |   **45.9**   |
> | SFT+GRPO             |  4.2/0.2   | 19.4  |   51.0   |  22.4   |   20.1   |     **19.6**     | 86.1  | 31.3  |   47.0   |    63.4    |   **57.0**   |
> | **AMPO (4LongCoTs)** |  9.4/1.7   | 26.9  |   59.6   |  37.9   |   25.7   |     **24.5**     | 88.0  | 29.8  |   52.8   |    68.9    |   **59.9**   |
> |                      |            |       |          |         |          |                  |       |       |          |            |              |
> | **Qwen-7B-Instruct** |  12.3/6.5  | 43.8  |   75.8   |  36.4   |   38.8   |     **35.6**     | 85.1  |  6.6  |   55.8   |    80.5    |   **57.0**   |
> | SFT                  |  12.7/14   | 41.0  |   76.0   |  30.5   |   36.4   |     **35.1**     | 80.0  | 17.2  |   45.2   |    82.9    |   **56.3**   |
> | GRPO                 |  11.3/9.9  | 46.6  |   76.8   |  34.6   |   37.6   |     **36.1**     | 92.1  |  5.1  |   58.7   |    74.4    |   **57.6**   |
> | SFT+GRPO             | 20.1/16.7  | 53.4  |   81.0   |  35.0   |   50.0   |     **42.7**     | 91.0  | 38.4  |   58.7   |    83.5    |   **67.9**   |
> | LUFFY                | 16.4/15.2  | 49.3  |   81.6   |  37.9   |   45.8   |     **41.0**     | 92.0  | 21.7  |   60.3   |    78.1    |   **63.0**   |
> | **AMPO (4LongCoTs)** |  14/14.1   | 48.0  |   80.8   |  39.3   |   45.9   |     **40.4**     | 92.7  | 40.4  |   59.6   |    81.1    |   **68.5**   |
> | AMPO (Only R1)       |  14.8/9.2  | 48.8  |   77.0   |  35.7   |   42.5   |     **38.0**     | 92.9  |  0.5  |   59.7   |    81.1    |   **58.6**   |
> | SFT+AMPO             |  19/18.3   | 53.2  |   82.0   |  37.5   |   49.5   |     **43.2**     | 90.7  | 38.4  |   56.7   |    82.3    |   **67.0**   |
>
> **Table 2: Comparison of Fixed $k_0$ vs. Dynamic $k_0$ Strategy**
> | $k_0$ Strategy         |   AIME 24/25    |   AMC    | MATH 500 | Minerva  | Olympiad | **Avg. Score** | **Avg. Length** |
> | :--------------------- | :-------------: | :------: | :------: | :------: | :------: | :------------: | :-------------: |
> | Fixed $k_0=1$          |   16.3 / 11.0   |   50.2   |   78.2   |   36.8   |   40.6   |      38.8      |      1104       |
> | Fixed $k_0=2$          |   17.5 / 12.2   |   52.6   |   80.6   |   37.1   |   46.1   |      41.0      |      2439       |
> | Fixed $k_0=3$          |   16.9 / 12.5   |   51.2   |   81.2   |   37.9   |   47.0   |      41.1      |      2103       |
> | Fixed $k_0=4$          |   16.6 / 15.3   |   51.3   |   82.2   |   38.2   |   45.3   |      41.5      |      1750       |
> | **Adaptive (Dynamic)** | **17.4 / 17.7** | **50.5** | **80.2** | **39.0** | **44.7** |    **41.6**    |    **1234**     |

---

### Meta-Review · Area_Chair_16Yr · 2026-01-13

**Summary:**

The primary concerns are limited conceptual novelty—reviewers noted AMPO combines existing mechanisms rather than introducing fundamentally new principles—and absence of theoretical foundation, as the paper lacks formal analysis (regret bounds, convergence guarantees) explaining when multi-teacher diversity provably outperforms single strong teachers. Additional issues include heuristic-based design choices (teacher selection, k₀) without rigorous justification, incomplete baselines for smaller models, and domain-specific focus primarily on mathematical reasoning with limited generalization evidence.

**Reviewer Concerns:**

The rebuttal successfully resolved experimental completeness issues by adding full baselines (SFT, SFT+GRPO) for smaller models, demonstrated multi-teacher superiority over single strong teacher under controlled conditions, validated the dynamic k₀ strategy, and extended evaluation to code generation (HumanEval). Technical clarifications regarding teacher log probabilities and notation ambiguities were also satisfactorily addressed.

The fundamental novelty critique remains unresolved—the framework is still viewed as an engineering combination rather than a conceptual breakthrough. The lack of theoretical foundation (formal analysis of when/why diversity helps) persists despite empirical evidence. The heuristic-based teacher selection and absence of principled diversity metrics were acknowledged but not fundamentally addressed, with authors deferring these to future work.

**Reviewer Scores:**

kg4R likely maintains 4-5, as computational cost and baseline concerns were addressed, but the requested k₀ guidance was only partially resolved. Lxdg had risen to 6, given their "neutral" comment—multi-teacher SFT clarification helped, but theoretical diversity metrics remain absent. dXdg probably stays 4, appreciating comprehensive responses on generalization/scalability, but core theoretical concerns unresolved. y1Nj likely remains 2, as fundamental novelty critique (AMPO as mere combination) and missing theoretical justification were not addressed at the conceptual level.

---

### Decision · Program_Chairs · 2026-01-26

Reject